# Matchmaker: Schema Matching with self-improving compositional LLM programs

## Abstract

Schema matching – the task of finding matches between attributes across disparate data sources with different tables and hierarchies – is critical for creating interoperable machine learning (ML)-ready data. Addressing this fundamental data-centric problem has wide implications, especially in domains like healthcare, finance and e-commerce — but also has the potential to benefit ML models more generally, by increasing the data available for ML model training. However, schema matching is a challenging ML task due to structural/hierarchical and semantic heterogeneity between different schemas. Previous ML approaches to automate schema matching have either required significant labeled data for model training, which is often unrealistic or suffer from poor zero-shot performance. To this end, we propose Matchmaker - a compositional language model program for schema matching, comprised of candidate generation, refinement and confidence scoring. Matchmaker also self-improves in a zero-shot manner without the need for labeled demonstrations via a novel optimization approach, which constructs synthetic in-context demonstrations to guide the language model's reasoning process. Empirically, we demonstrate on real-world medical schema matching benchmarks that Matchmaker outperforms previous ML-based approaches, highlighting its potential to accelerate data integration and interoperability of ML-ready data.

## 1 Introduction

The success of machine learning (ML) models hinges on a critical yet often overlooked challenge: access to large, integrated and interoperable datasets (Jain et al., 2020; Gupta et al., 2021; Renggli et al., 2021; Sambasivan et al., 2021). Although well-structured and uniform datasets like those on Kaggle are commonly assumed as the norm, such data is a rare luxury in practice. In real-world scenarios, tabular data often exists in heterogeneous and disparate databases with diverse formats, schemas, and terminologies, requiring harmonization to make the data "ML-ready" and interoperable. The heterogeneity of databases presents three critical issues for ML: (1) data harmonization and integration is an arduous task. Hence, researchers often limit the features/covariates used for model training to a smaller, often common, set of features (Avati et al., 2021; Si et al., 2021; Rajkomar et al., 2018), thereby limiting the potential performance of their ML models; (2) even if all the features are used, the lack of data interoperability means limited external validation of ML models (Balch et al., 2023; Lehne et al., 2019; Williams et al., 2022; Tiwari et al., 2020; Colubri et al., 2019), which can undermine the credibility and utility of the ML models; and (3) missed opportunities for insights on larger harmonized datasets (e.g., larger patient populations), which may not be apparent when analyzing data sources independently.

Schema matching is a critical first step in data harmonization, aiming to establish correspondences between attributes (i.e., features/covariates) measured across different data sources. Once matched, these correspondences can help harmonize data from disparate sources into a cohesive, ML-ready format. To understand the concept of schema matching, let us unpack the components of a schema. A schema defines how data is organized in a database, comprising different tables (collections of related data entries) and columns (also known as "attributes" or "features") that represent specific data fields. Importantly, schemas go beyond simple tabular data commonly found in CSV files, as they capture the hierarchical structure and relationships between different tables and their attributes. For example, in healthcare, schemas from different hospitals may have varying tables and attributes representing patient information, lab measurements, diagnoses and treatments, with complex relationships and

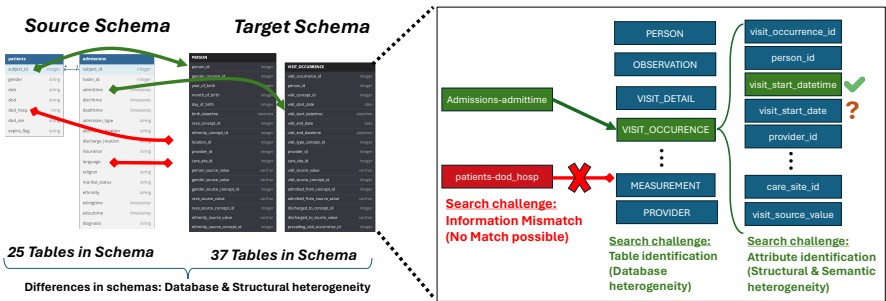

Figure 1: Example showing the complexity of schema matching due to the multi-faceted challenges: **Database heterogeneity (green arrows):** Identifying the correct target table is the first step, as each schema has a different number of tables, the corresponding information may be distributed differently across tables in each schema. **Structural heterogeneity (green arrows):** Once the appropriate table is found, matching attributes is complicated by differences in schema architectures, hierarchies, and granularity. **Textual heterogeneity (green arrows):** Ambiguity in matching when attributes have the same names but different meanings, or different names with the same meaning. **Information mismatch (red arrows):** Some attributes in one schema may lack a corresponding match in the other schema, adding to the complexity of the matching process.

hierarchies connecting the tables. Consequently, schema matching involves analyzing the context of attributes within the schema hierarchy to establish meaningful mappings that preserve the intended semantics and relationships. It goes beyond simple one-to-one column matching, considering not only the attribute itself but also the hierarchical structure and relationships between tables defined by the schema. Notably, schema matching does not assume access to raw data, relying on only attribute names, descriptions and metadata (e.g., in healthcare, patient data cannot be queried or accessed directly due to privacy concerns or regulations (Zhang et al., 2021)).

The importance and value of schema matching cannot be overstated, as integrating data from various data sources such as different regions, organizations or applications is vital in healthcare but also in finance and e-commerce (Sheetrit et al., 2024; Zhang et al., 2021; El Haddadi et al., 2024). Schema matching is also generally valuable to *anyone* working on ML, as a step toward increasing the training and validation data available to the ML community for model training. e.g, in healthcare, integrating data from multiple hospitals can lead to more comprehensive datasets to train more performant ML prognostic models. Similarly, in e-commerce, combining diverse customer data from various platforms can enable more accurate ML models built on customer data.

Unfortunately, prior ML approaches for "automated" schema matching often require extensive labeled data (Li et al., 2020; Zhang et al., 2021), which is often costly and time consuming to acquire, making these methods impractical for real-world use. Although LLM-based methods (Narayan et al., 2022; Mirchandani et al., 2023) have attempted to address this, they have poor zero-shot performance and poor scalability in terms of the number of LLM calls. These limitations have hindered the adoption of ML for schema matching, meaning schema matching is still a largely manual and time-consuming task. To highlight the need for automated and better performing ML schema matching, in the healthcare domain, it took 500 hours for two experts to map the schemas between the MIMIC database and the OMOP common data model (Paris et al., 2021), demonstrating the substantial and non-trivial effort required.

Despite the need, schema matching is a challenging ML task, as shown in Fig. 1, as without access to the raw data, schema matching methods must rely only on the attribute names and other metadata to infer correspondences between attributes across schemas. This requires reasoning about various challenges, namely: ▶ **Semantic heterogeneity:** ambiguous potential mappings, where attributes across schemas might have the same name but different meanings, or different names but the same meaning. ▶ **Structural heterogeneity:** schemas that have varied architectures, hierarchies, and representational granularity. ▶ **Database heterogeneity:** differences in the number and organization of tables across schemas. e.g. source schema table information may be represented across multiple target schema tables. Hence, it is non-trivial to identify the appropriate table for an attribute. ▶ **Information mismatch:** Information may be contained in one schema, but not in another schema. Hence, reasoning about "no possible match" is as important as reasoning about a possible match.

These issues make schema matching a challenging task that cannot be solved by simple methods such as semantic similarity alone (see Fig. 2). To this end, we introduce *Matchmaker*, a self-improving compositional language model program for schema matching. Matchmaker leverages the reasoning capabilities of large language models (LLMs) via a compositional language model program with multi-stage LLM calls that comprise candidate generation, refinement, and confidence scoring (see Appendix C for examples of this process). Matchmaker also *self-improves* without labeled data (zero-shot), via a novel optimization process using *synthetic in-context examples* for the different stages of the language model program. Matchmaker makes the following contributions:

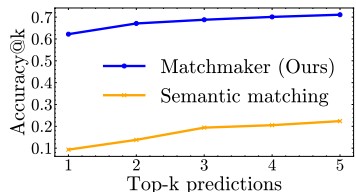

Figure 2: Example result shows semantic similarity alone cannot solve schema matching, with low accuracy@k, compared to Matchmaker.

**Contributions:** ① We address recent calls to develop ML methods for data harmonization/interoperability (Balagopalan et al., 2024; Gilbert et al., 2024). ② We introduce a novel formulation of schema matching as information retrieval rather than binary classification. ③ We propose Matchmaker, a novel compositional language model program to address the complexities of schema matching. ④ We introduce a scoring mechanism allowing for human-in-the-loop deferral not possible with prior methods. ⑤ We introduce a novel optimization mechanism allowing Matchmaker to self-improve in a zero-shot manner via synthetic in-context examples that guide Matchmaker's reasoning process. ⑥ We empirically demonstrate that Matchmaker outperforms different schema matching baselines on real-world schema matching benchmarks, along with showing the value of our self-improvement mechanism and how Matchmaker can be used with a human-in-the-loop.

## 2 RELATED WORK

This work engages with literature on schema matching (see Fig. 3) and contributes to data-centric AI.

**Schema matching.** Previous ML-based schema matching approaches have shown promise, but suffer from limitations that hinder their practical applicability. Early works (Mudgal et al., 2018; Shraga et al., 2020; Li et al., 2020) computed similarity scores between schemas (Do & Rahm, 2002; Gal, 2011), but focused on the simpler entity matching task (matching items within columns) rather than the more complex schema matching problem. Recent methods like SMAT (Zhang et al., 2021) applied deep learning (i.e. attention) to tackle full schema matching. However, they require extensive labeled matches (over 50%), rendering them impractical for real-world environments where labeled data is scarce or expensive to obtain, often requiring domain experts to annotate.

To reduce the need for labels, LLMs have been applied to schema matching (Zhang et al., 2023a; Narayan et al., 2022; Zhang et al., 2023b). However, methods like LLM-DP using pre-trained LLMs (Zhang et al., 2023a; Narayan et al., 2022) have demonstrated poor zero-shot performance (see Sec. 5). Performance improvements were obtained with human-labeled examples of ±500 examples, from which in-context examples are selected. However, reliance on human labeling is often unrealistic, limiting applicability. Interestingly, even LLMs such as Jellyfish (Zhang et al., 2023b), which are fine-tuned for schema matching on task datasets, have shown poor matching performance. Beyond matching performance, both LLM and supervised methods (e.g. SMAT (Zhang et al., 2021)), formulate schema matching as a binary classification task over the full Cartesian product of source and target schema attributes. e.g. for each pair of source-target attributes, the LLM is prompted to provide a label of Yes/No for the match (i.e. Is attribute A related to Attribute B? yes/no). The result is poor scalability which is computationally expensive for large schemas and costly due to the large number of LLM calls, hindering real-world applicability. We compare LLM calls in Appendix D.1.

The closest work to ours is ReMatch (Sheetrit et al., 2024), which uses retrieval to find semantically similar candidate matches, thus reducing the search space. It then prompts an LLM to match a source schema attribute with retrieved target schema candidates. However, ReMatch relies solely on semantic matching, which we empirically demonstrate in Sec. 5 does not suffice for real-world schemas. Our approach Matchmaker diverges from ReMatch along three dimensions (see Table 1): (1) *System*: ReMatch uses a single LLM call, while Matchmaker decomposes the task into a multi-stage compositional LLM program with iterative reasoning steps. (2) *Candidate generation*: ReMatch relies solely on semantic retrieval, while Matchmaker incorporates *diverse* candidate generation sources, including retrieval for semantic candidates and an LLM-driven contextual reasoning candidates. (3) *Optimization*: ReMatch has a fixed/static LLM prompt template, while Matchmaker is an LLM program where we dynamically optimize the prompts via synthetic in-context examples.

Table 1: Difference between Matchmaker & the closest work ReMatch along multiple dimensions.

| Feature | ReMatch | Matchmaker (Ours) |
|---|---|---|
| **System/Approach** | Single-step process $R : A_s \times A_t \to \{0, 1\}$ | Multi-step compositional LM program $L = \{l_1, l_2, \ldots, l_n\}$ (Sec 4) |
| **Candidate Generation** | Semantic retrieval only $C_s = g(A_{si}, A_t)$ | Semantic + Reasoning-based $C = C_R \cup C_s$ (Sec 4.2) |
| **Reasoning Mechanism** | Limited to ranking stage | Chain-of-Thought prompting throughout |
| **Ranking** | Single LLM call for binary decision $R(A_{si}, A_{tj}) \in \{0, 1\}$ | LLM-based confidence scoring with MCQ format: $r : (A_s \times D_s) \times (A_t \times D_t) \to \mathbb{R}$ allowing for uncertainty deferral (Sec 4.3) |
| **Self-improvement/Optimization** | None | Zero-shot with synthetic examples: $E : (e_i, L(e_i)) \to \mathbb{R}$ (Sec 4.4) |
| **LLM Prompts** | Static | Dynamic |

**Data-Centric AI.** Data-centric AI is an area of growing importance in the ML community. It aims to systematically improve data quality for ML (Zha et al., 2023; Whang et al., 2023) through methods such as sample selection and valuation (Seedat et al., 2023; Jiang et al., 2023) of pre-existing integrated datasets. This work addresses a fundamental upstream problem: schema matching which enables the creation of harmonized datasets. Consequently, schema matching is a contribution to data-centric AI literature by tackling a critical issue that precedes and supports existing approaches to enhance data quality for ML.

## 3 SCHEMA MATCHING

### 3.1 PRELIMINARIES.

Consider the schema matching task, where the goal is to map attributes from a source schema $(S_s)$ to a target schema $(S_t)$. Each schema $S$ is defined as a collection of tables $\mathcal{T} = \{T_1, T_2, \ldots, T_m\}$. Each table $T_i$ contains a set of attributes $\mathcal{A}_i = \{A_{i1}, A_{i2}, \ldots, A_{ik}\}$. Additionally, each table $T_i$ is associated with metadata $m_i$ describing the purpose and content of the table. Similarly, each attribute $A_{ij}$ is associated with a description $d_{ij}$, which includes information describing the attribute, its data type and relational context. These descriptions and data types offer key contextual information to aid in the matching process.

The schema matching task, defined below, aims to find matches between attributes across different schemas, accounting for their structural hierarchies, interrelationships and constraints.. Recall that schema matching operates solely on schema-level information (attributes and metadata), without having access to the raw data. This adds to the complexity, as matching must be performed without the benefit of analyzing the actual data values.

**Definition 1** (Schema Matching). *The goal of schema matching is to find a mapping function* $f : \mathcal{A}_s \to \mathcal{A}_t \cup \{\varnothing\}$ *that correctly assigns each attribute of the source schema $S_s$ to a corresponding attribute in the target schema $S_t$ or to the empty set $\varnothing$, indicating no possible match.*

### 3.2 SCHEMA MATCHING AS INFORMATION RETRIEVAL.

As outlined in Sec. 2, schema matching is often formulated as a supervised binary classification problem (match/no match) over the entire Cartesian product of source and target schema attributes. Beyond the computational side, this formulation has several drawbacks: ▶ **Labeling Cost:** It requires manual annotation of attribute pairs by domain experts, which is time-consuming and costly. ▶ **Class Imbalance:** The prevalence of non-matching attribute pairs significantly outnumbers matching pairs, resulting in severe class imbalance. ▶ **Lack of Ranking:** It does not yield a ranked list of candidate matches, which is critical for human review if multiple possible matches exist.

▶ **1. Candidate generation**: For each source query attribute $A_{si} \in \mathcal{A}_s$ from the source schema $S_s$, we generate a set of potential matches from the target schema $S_t$. Let $C_i \subseteq \mathcal{A}_t$ be the set of candidate target matches for query attribute $A_{si}$. The candidate generation process is defined as a function $g : \mathcal{A}_s \times \mathcal{A}_t \to \mathcal{P}(\mathcal{A}_t)$, where $\mathcal{P}(\mathcal{A}_t)$ denotes the power set of $\mathcal{A}_t$, such that $C_i = g(A_{si}, \mathcal{A}_t)$.

▶ **2. Ranking**: We rank the candidates based on their relevance to the query attribute. We define a ranking function $r : (\mathcal{A}_s \times \mathcal{D}_s) \times (\mathcal{A}_t \times \mathcal{D}_t) \to \mathbb{R}$, where $\mathcal{D}_s$ and $\mathcal{D}_t$ represent the contextual information associated with attributes in $\mathcal{A}_s$ and $\mathcal{A}_t$, respectively. For each source attribute $A_{si} \in \mathcal{A}_s$ and its associated contextual information $d_{si} \in \mathcal{D}_s$, the ranking function $r$ assigns a relevance score to each candidate attribute $A_{tj} \in C_i \subseteq \mathcal{A}_t$ and its associated contextual information $d_{tj} \in \mathcal{D}_t$:

$$r((A_{si}, d_{si}), (A_{tj}, d_{tj})) > r((A_{si}, d_{si}), (A_{tk}, d_{tk})) \Leftrightarrow A_{tj} \text{ is more relevant to } A_{si} \text{ than } A_{tk}.$$

Figure 3: Conceptual comparison of different schema matching approaches. **(A)** Supervised Matching (Zhang et al., 2021) employs a trained neural network (e.g., a transformer) to predict binary match/no-match labels across all attribute pairs, scaling as $\mathcal{O}(n)^2$ and requiring labeled data, thus unsuitable for zero-shot. **(B)** LLM-Prompting (Narayan et al., 2022; Zhang et al., 2023a) uses a frozen language model (e.g., GPT-4) for the same task, with similar scalability. Alternatively, (Zhang et al., 2023b) fine-tunes the LLM, which requires labeled data. **(C)** RAG-Based (Sheetrit et al., 2024) improves scalability by retrieving candidates from a vector database and using a frozen LLM to select matches, but its effectiveness is limited to semantically similar options. **(D)** Matchmaker (Ours) performs schema matching via a self-improving, compositional language model program that enables enhanced reasoning. The program includes both retrieval and reasoning-based candidate generation with refinement and confidence scoring, allowing for more accurate ranking. The program is optimized using synthetic in-context examples in the LLM prompts.

The mapping function $f$ can then be defined as follows:

$$f(A_{si}) = \begin{cases} \arg\max_{A_{tj} \in C_i} r((A_{si}, d_{si}), (A_{tj}, d_{tj})), & \text{if } \max_{A_{tj} \in C_i} r((A_{si}, d_{si}), (A_{tj}, d_{tj})) \geq \tau \\ \varnothing, & \text{otherwise} \end{cases}$$

where $\tau$ is a relevance threshold and $f$ assigns the query attribute $A_{si}$ to the candidate attribute $A_{tj}$ with the highest relevance score. Conversely, we may assign $\varnothing$, indicating no match — accounting for the fact that not all source attributes may have a possible match in the target schema. Further details can be found in Appendix A.5

## 4 MATCHMAKER: LLM-BASED SCHEMA MATCHING

We propose Matchmaker, a self-improving compositional language model (LM) program for schema matching (see Fig. 3), defined as a three-step LM program. For further details see Appendix A.2.

1. **Multi-vector documents** (Sec. 4.1): Creation of multi-vector documents from the target schema to facilitate semantic candidate retrieval of potential target attribute matches.
2. **Candidate generation** (Sec. 4.2): Employing two types of candidate generation: semantic retrieval and reasoning-based. The candidates are then refined into a smaller candidate set to evaluate.
3. **Confidence scoring** (Sec. 4.3): match confidence of a candidate target attribute to a query attribute.

> 💡 *Steps 1-3 define the unoptimized Matchmaker program. Finally, a key aspect of Matchmaker is our zero-shot optimization via synthetic in-context examples to improve performance (Sect. 4.4).*

**Why LLMs for schema matching?** Large Language Models (LLMs) form the foundation of Matchmaker, serving as key components within a compositional program comprised of multiple language model calls. Specifically, LLMs exhibit several appealing properties and capabilities for schema matching: ▶ **Contextual understanding**: LLMs have been pretrained on vast corpora of information, equipping them with extensive prior knowledge spanning different contexts and settings (Chowdhery et al., 2022; Singhal et al., 2023). This contextual understanding enables LLMs to effectively reason about schema hierarchies and identify potential matches. ▶ **Hypothesis proposers**: LLMs have been shown to be "phenomenal hypothesis proposers" (Qiu et al., 2023), making them particularly useful for candidate generation tasks. ▶ **Capable rankers**: LLMs have been shown to be highly capable at relevance ranking; assessing the suitability of candidates given a query and a set

of options (Zhuang et al., 2023; Hou et al., 2024), especially "when ranking candidates retrieved by multiple candidate generators" (Hou et al., 2024).

**Defining a compositional LM program.** A compositional language model program, denoted as $\mathcal{L}$, is a multi-stage pipeline consisting of multiple LLM calls, i.e., $\mathcal{L} = \{l_1, l_2, \ldots, l_n\}$, where $l_i : (s, k_s) \to \mathcal{Y}$ represents a specific LLM call taking as input a prompt string $s$ and in-context examples $k_s$ (which could be $\varnothing$). In the following sections (Secs. 4.1-4.3), we define the different components of $\mathcal{L}$ specific to Matchmaker. Finally, we describe our optimization process (Sec. 4.4).

### 4.1 MULTI-VECTOR DOCUMENTS (STEP 1)

To efficiently retrieve semantically similar candidates from the target schema, we build a vector database that encodes target schema attributes. We begin by representing the target schema as a collection of structured documents. Specifically, for each table $T$ in the target schema $S_t$, we create a document for each table consisting of the attribute names and append the attribute's textual description and data type, providing contextual information about each attribute. The metadata of each document includes the description of the table itself.

Unlike conventional approaches that encode each document as a single high-dimensional vector, Matchmaker utilizes multi-vector representations. Specifically, we use ColBERT-v2 (Santhanam et al., 2022) to encode the document chunks, producing an embedding per token (i.e., token-level dense vector), capturing token-level interactions — as it has been shown to provide improvements in expressivity (Thakur et al., 2021; Lee et al., 2024) and out-of-domain performance (Santhanam et al., 2022). The following section explains how semantically similar candidates are retrieved using this multi-vector representation.

### 4.2 DIVERSE CANDIDATE GENERATION (STEP 2)

To narrow down the search space, Matchmaker identifies a subset of candidate attributes from the target schema that are likely matches for a query attribute $q_i \in A_s$ from the source schema. We draw inspiration from (Hou et al., 2024), which demonstrates that LLM ranking performance improves "'when ranking candidates are retrieved by multiple candidate generators." Hence, while semantic candidates are commonly used, Matchmaker goes beyond and employs two distinct types of candidate generation: (i) Semantic retrieval candidates retrieved from the vector database, and (ii) Reasoning-based candidates using a language model. This is then followed by a candidate refinement step. We outline each type of candidate generation applicable to a given query attribute $q_i \in A_s$.

**(i) Semantic retrieval candidates.** Given query $q_i$, we encode it using ColBERT-V2, producing a multi-vector query embedding. Matchmaker then uses this query embedding to retrieve the top-k matching target schema attributes in the vector database. The top-k semantically similar candidates are denoted as $\mathcal{C}_s$. Similarity is computed using a late-interaction approach (Khattab & Zaharia, 2020), though a Maxsim operator which identifies the highest similarity scores for the query tokens, and these scores are aggregated to generate a relevance score for that document. All documents are then ranked based on their overall relevance scores. The top-k documents, which contain the most semantically similar attributes to the query, are retrieved as matches.

**(ii) Reasoning-based candidates.** To complement semantic matches, Matchmaker generates reasoning-based candidates using a candidate reasoner LLM denoted as $l_c : (q_i, \mathcal{A}_t) \to \mathcal{C}_R$, where $q_i$ is the i-th query, $\mathcal{A}_t$ is the set of all target attributes and $\mathcal{C}_R$ is a reasoning-based candidate set. Matchmaker employs Chain of Thought (CoT) prompting (Wei et al., 2022) to reason about the target attributes $\mathcal{A}_t$ given the context of the schema hierarchy, descriptions and data types — generating the most likely and relevant target schema candidate matches for each query $q_i$. This metadata allows the LLM to reason about the schema structure and relationships beyond just attribute names.

**Refinement.** At this stage, the set of candidates is $\mathcal{C} = \mathcal{C}_R \cup \mathcal{C}_s$. Matchmaker then refines this set by selecting the most relevant candidates for each query attribute, resulting in a smaller, prioritized candidate set $\mathcal{C}^*$ to score and rank. Candidate refinement is achieved with a refiner LLM using CoT, denoted as $l_r : s \to \mathcal{C}^*$, where $s = (\mathcal{C}, q_i)$ and $q_i$ is the i-th source query.

### 4.3 CONFIDENCE SCORING (STEP 3)

The refined set of candidates, $\mathcal{C}^*$ remains unordered. Hence, this step aims to obtain confidence scores to rank the candidates but also gauge the certainty of each match, recognizing that sometimes

no suitable source-to-target attribute match exists, which requires the system to abstain from making a match. While language models may not be well-calibrated at the sequence level, recent research has shown that they exhibit better calibration at the token level (Ren et al., 2023), a feature notably beneficial in multiple-choice question (MCQ) tasks (Kadavath et al., 2022). Leveraging this insight, Matchmaker structures the candidate scoring task as an MCQ format, labeling each candidate in $\mathcal{C}^*$ for query $q_i$ as options *(A), (B), (C), etc.* Additionally, to account for none of the target candidates being a good match or there might be no possible match in the target schema, Matchmaker includes an abstain option by adding "NONE of the above" as a choice. This ensures that the LLM is not forced to select a candidate when there is no suitable match (Ren et al., 2023; Ding et al., 2023).

Matchmaker finally performs candidate ranking, where it is common to evaluate each candidate individually (Hu et al., 2024; Wang et al., 2023a; Zheng et al., 2023). Confidence scores are obtained by prompting the LLM to reason about the relevance of each candidate $c_i \in \mathcal{C}^*$ to the given query $q_i$. Furthermore, prior work has shown that LLMs can provide good uncertainty at token-level (Kadavath et al., 2022) like in our MCQ, which is achievable via prompting (Tian et al., 2023). Consequently, Matchmaker elicits a confidence score by prompting the LLM to provide a value between 0 and 100, indicating the relevance of a match. The confidence scores are used to rerank candidates or, if "None of the above" receives the highest score, return an empty list (i.e. no suitable matches for the query).

## 4.4 Self-improvement: Zero-shot optimization w/ synthetic in-context examples

Matchmaker optimizes the language model program $\mathcal{L}$ by leveraging the few-shot learning capabilities of LLMs (Brown et al., 2020; Agarwal et al., 2024; Dong et al., 2022). This is achieved by selecting input-output demonstrations (i.e. in-context examples). In Sec. 5, we contrast this with an alternative self-improvement method via self-reflection. However, selecting in-context examples is non-trivial for schema matching for two reasons.

(i) **Lack of labeled demonstrations:** We do not have access to labeled input-output demonstrations from which to select in-context examples. To overcome this challenge, we use the unlabeled schemas to create a "evaluation" set $\mathcal{D}_{eval} = \{e_1, e_2, \ldots, e_m\}$, made up of different types of source queries. Specifically, we identify "easy queries" where the top-n (n=5) target schema semantic matches have a similarity score $> 0.95$, and "challenging queries" with the lowest semantic matches.

(ii) **Lack of an evaluator:** To assess Matchmaker's capabilities on the evaluation set and guide the optimization process, we need a validation metric. Since no validator is readily available, we propose to use an evaluator LLM, $\mathcal{E} : (e_i, \mathcal{L}(e_i)) \rightarrow \mathbb{R}$, that employs chain of thought (Wei et al., 2022) to score the relevance (from 0-5) of matches obtained from $\mathcal{L}$ when evaluated on examples from $\mathcal{D}_{eval}$.

---

**Algorithm 1** Optimize LM program $\mathcal{L}$

---

1: **Input:** Set of evaluation queries $\mathcal{D}_{eval} = e_1, e_2, \ldots, e_n$
2: **Output:** Set of top $n$ demonstrations $D_{demo}$
3: **for** each input $e_i \in \mathcal{D}_{eval}$ **do**
4: $\quad \hat{y}_i, trace_i \leftarrow \mathcal{L}(e_i)$ $\qquad\qquad$ ▷ Teacher $\mathcal{L}$ predicts, storing outputs and intermediate traces
5: $\quad s_i \leftarrow \mathcal{E}(e_i, \hat{y}_i)$ $\qquad\qquad\qquad\qquad\qquad\qquad\qquad\qquad\qquad$ ▷ Evaluation score
6: $\quad D_{demo} \leftarrow D_{demo} \cup (e_i, trace_i, \hat{y}_i, s_i)$
7: **end for**
8: Sort $D_{demo}$ by score
9: **return** $D_{demo}[0 : n]$ $\qquad\qquad\qquad\qquad\qquad\qquad\qquad\qquad\qquad$ ▷ Select top $n$

---

**Zero-shot optimization w/ synthetic in-context examples.** To optimize our multi-stage language model program, we aim to select in-context examples for each component in $\mathcal{L}$. However, in-context demonstrations for the intermediate stages are typically unavailable.

To address this, we simulate *traces* by running $\mathcal{L}$ on the evaluation examples $e_i \in \mathcal{D}_{eval}$. A trace captures the intermediate input-output pairs of each component in $\mathcal{L}$ during the execution of $\mathcal{L}$ on a given example. The evaluator $\mathcal{E}$ then scores the final output, assessing Matchmaker's ($\mathcal{L}$) overall performance on each example. We then adopt a bootstrapping process (Khattab et al., 2023) that selects the intermediate input-output pairs from the traces that produced the highest evaluation scores as synthetic in-context examples for each component of $\mathcal{L}$. In other words, we use the input-output pairs generated by Matchmaker itself (which resulted in good evaluation performance) as synthetic

in-context examples to guide the LLM reasoning. This allows us to improve the program in a zero-shot manner, without relying on actual labeled data. Algorithm 3 provides an overview of the process. We refer to $\mathcal{L}$ with the systematically selected in-context examples as Matchmaker (Optimized).

## 5 EXPERIMENTS

We now empirically investigate multiple aspects of Matchmaker. For qualitative examples that illustrate Matchmaker's application, refer to Appendix C.

| Sec. | Experiment | Goal |
|------|-----------|------|
| 5.1 | Overall performance | *Performance of Matchmaker vs schema matching benchmarks* |
| 5.2 | Self-improvement | *Performance of Matchmaker: optimized vs unoptimized vs random ICL vs self-reflection improvement* |
| 5.3 | Source of gain | *Ablation to understand Matchmakers candidate generation* |
| 5.4 | Matchmaker in practice | *Using Matchmaker with humans: uncertainty deferral and remedial action* |

**Setup.** We conduct experiments on the MIMIC-OMOP and Synthea-OMOP datasets, which are the standard benchmark datasets used in prior schema matching works (Sheetrit et al., 2024; Zhang et al., 2023b; Narayan et al., 2022; Zhang et al., 2023a; 2021). These datasets are real-world healthcare schema matching datasets and have been widely adopted due to their complexity and their reflection of real-world schema matching challenges. Additionally, complex, real-world schema matching datasets are rare and difficult to obtain, as annotating them requires extensive domain expertise (e.g., 500 hours for MIMIC-OMOP), making them invaluable test beds for schema matching algorithms. An overview of the datasets is provided in Appendix B, along with further experimental details.

**Metrics.** We evaluate schema matching performance using accuracy@k used in (Sheetrit et al., 2024) and is commonly used in information retrieval. Besides, ReMatch the other baselines treat schema matching as a binary classification using F1-score as the metric. In our setting of m:1 matching (i.e. one match for each query), accuracy@1 is equivalent to F1-score, precision and recall, if the label is assigned via $argmax$. For details see Appendix A.8. Hence, we report accuracy@1 for all other baselines for comparison to retrieval based approaches. Unless otherwise stated, metrics are averaged over 5 seeds (with standard deviation).

### 5.1 SCHEMA MATCHING PERFORMANCE: DOES IT WORK?

Matchmaker's performance is compared to diverse schema-matching baselines (refer to Sec. 2). These include (i) LLM-based methods such as ReMatch and LLM-DP, (ii) the state-of-the-art non-LLM supervised model, SMAT, and (iii) Jellyfish, an LLM specifically fine-tuned for data preprocessing tasks, including schema matching. While Jellyfish is fine-tuned using the same MIMIC and Synthea datasets, giving it an advantage, we include it as a baseline to highlight Matchmaker's zero-shot performance using a general-purpose LLM. This comparison spans general-purpose LLMs, traditional supervised approaches, and task-specific fine-tuned models. All LLM baselines use GPT-4 (0613) (OpenAI, 2023) as the backbone for fair comparison to the original works and to isolate the gains of the system which aren't tied to the LLM. Other LLM backbone results are found in Appendix D, showing Matchmaker's gain isn't due to the LLM alone.

Table 2: Comparison of schema matching performance of different baselines.

| | | Matchmaker | ReMatch [1] | JellyFish-13b | Jellyfish-7b | LLM-DP | SMAT (20-80) | SMAT (50-50) |
|---|---|---|---|---|---|---|---|---|
| MIMIC | acc@1 | **62.20 ± 2.40** | 42.50 | 15.36 ± 5.00 | 14.25 ± 3.00 | 29.59 ± 2.00 | 6.05 ± 5.00 | 10.85 ± 6.00 |
| | acc@3 | **68.80 ± 2.00** | 63.80 | N.A. | N.A. | N.A. | N.A. | N.A. |
| | acc@5 | 71.10 ± 2.00 | **72.90** | N.A. | N.A. | N.A. | N.A. | N.A. |
| Synthea | acc@1 | **70.20 ± 1.70** | 50.50 | 35.17 ± 3.90 | 31.52 ± 1.70 | 41.44 ± 5.40 | 36.23 ± 3.30 | 44.88 ± 2.60 |
| | acc@3 | **78.60 ± 2.50** | 58.10 | N.A. | N.A. | N.A. | N.A. | N.A. |
| | acc@5 | **80.90 ± 1.10** | 74.30 | N.A. | N.A. | N.A. | N.A. | N.A. |

**Matchmaker has the best overall performance.** Matchmaker consistently outperforms baselines, across all settings, as shown in Table 2. Importantly, we find the largest performance gains *(+-20%)* for accuracy@1. This is a desirable property, as it suggests a better ranking of matches. Moreover, a higher accuracy at low $k$ values enables the use of smaller prediction sets, reducing the human effort required to select the final best target attribute match for a given source attribute query.

---

[1] ReMatch code implementation not available, hence we report the best accuracy@k values with the retrieval step as in (Sheetrit et al., 2024). Appendix D shows results for a re-implementation of ReMatch.

**Formulation as information retrieval outperforms binary classification.** A key insight from our experiments is that information retrieval-based approaches (Matchmaker and ReMatch) perform substantially better for accuracy@1 compared to the other binary classification-based approaches, which evaluate the full Cartesian product of attributes. This performance gap can be attributed to the smaller search space of the information retrieval formulation. Notably, Matchmaker and ReMatch are evaluated on all mappings, including matches and nulls ("No possible match"), whereas binary classification methods consider a simpler problem by only evaluating true matches.

## 5.2 MATCHMAKER SELF-IMPROVEMENT ANALYSIS

Matchmaker self-improves its language model program in a zero-shot manner (no labeled examples) via an optimization process using synthetic in-context examples (Sec. 4.4). We evaluate the performance of Matchmaker (Optimized) to three alternatives to disentangle the value of our in-context example selection mechanism: (1) Matchmaker (Vanilla), which is the vanilla language model program without in-context examples, (2) Matchmaker (Random): random selection of in-context examples rather than our optimized/systematic selection of in-context examples and (3) Matchmaker (Self-Reflection), which employs a self-reflection or self-refinement mechanism (Pan et al., 2023; Madaan et al., 2024) as an alternative self-improvement approach. i.e. the LLM iteratively self-corrects through feedback and has been used for various LLM tasks to improve performance.

The results in Table 3 illustrate the following: ▶ Matchmaker (Optimized) achieves significant performance gains compared to Matchmaker (Unoptimized), particularly at low $k$ values (+-5% improvement for acc@1). This finding highlights the value of the synthetic in-context examples and the potential for zero-shot self-improvement, even in the absence of labeled data or well-defined evaluation metrics. ▶ Matchmaker (Optimized) outperforms Matchmaker (Random), confirming that our systematic selection of in-context samples is the key driver of performance gains, rather than the mere inclusion of *any* in-context examples. ▶ Matchmaker (Optimized) which uses an LLM evaluator to score demonstration examples directly, provides better performance gains compared to the self-reflection approach, where an LLM simply self-refines along the pipeline. This underscores the importance of input-output demonstrations for Matchmaker, especially considering the multi-stage nature of the program, where the outputs of earlier components affect later components.

Table 3: Comparison of different Matchmaker self-improvement mechanisms, showing the value of our systematic selection of in-context samples vs random selection, vanilla or improvement via self-reflection.

| | | Matchmaker (Systematic - Full) | Matchmaker (Random) | Matchmaker (Vanilla) | Matchmaker (Self-reflection) |
|---|---|---|---|---|---|
| MIMIC | acc@1 | **62.20 ± 2.40** | 55.36 ± 2.15 | 57.90 ± 1.20 | 57.10 ± 0.60 |
| | acc@3 | **68.80 ± 2.00** | 62.74 ± 4.50 | 66.40 ± 0.60 | 66.60 ± 1.00 |
| | acc@5 | **71.10 ± 2.00** | 65.00 ± 6.42 | 70.20 ± 0.70 | 70.60 ± 0.50 |
| Synthea | acc@1 | **70.20 ± 1.70** | 67.76 ± 1.38 | 65.40 ± 0.90 | 67.80 ± 1.40 |
| | acc@3 | **78.60 ± 2.50** | 76.19 ± 5.28 | 78.20 ± 0.60 | 75.90 ± 0.70 |
| | acc@5 | 80.90 ± 1.10 | 77.66 ± 5.07 | **83.20 ± 1.10** | 81.10 ± 1.90 |

## 5.3 SOURCE OF GAIN ABLATION: WHY DOES IT WORK?

Matchmaker's performance relies on the generated candidate matches. Given its strong performance compared to baselines, we investigate which candidate generation approach contributes most to Matchmaker's success. To disentangle the role of each candidate generation method, we assess Matchmaker with (1) reasoning-based candidates from the LLM only (`Matchmaker_reasoning_only`) and (2) semantic candidates via retrieval only (`Matchmaker_semantic_only`).

The results in Table 4 show that reasoning-based candidates outperform semantic retrieval-based candidates. This finding suggests that LLM reasoning over the database hierarchy and data types produces better candidates than semantic matches that do not consider hierarchical relationships. In some cases (e.g., Synthea acc@1), the inclusion of retrieval-based candidates harms performance. However, the overall results indicate that Matchmaker benefits from both candidate generation approaches, with reasoning-based candidates providing greater value. These results highlights the value of diverse candidate generation mechanisms to enhance Matchmaker's overall performance.

Table 4: Understanding the impact of different candidate generation approaches on Matchmaker.

| | | Matchmaker | Matchmaker_reasoning_only | Matchmaker_semantic_only |
|---|---|---|---|---|
| MIMIC | acc@1 | $62.20 \pm 2.50$ | $61.60 \pm 1.50$ | $60.20 \pm 2.20$ |
| | acc@3 | $68.80 \pm 2.00$ | $68.70 \pm 1.60$ | $64.50 \pm 2.80$ |
| | acc@5 | $71.10 \pm 2.00$ | $70.40 \pm 1.00$ | $67.10 \pm 3.10$ |
| Synthea | acc@1 | $70.20 \pm 1.70$ | $73.00 \pm 1.90$ | $63.10 \pm 0.70$ |
| | acc@3 | $78.60 \pm 2.50$ | $78.50 \pm 1.50$ | $77.40 \pm 0.90$ |
| | acc@5 | $80.90 \pm 1.10$ | $79.40 \pm 0.30$ | $80.20 \pm 0.40$ |

### 5.4 MATCHMAKER IN PRACTICE: HUMAN-IN-THE-LOOP DEFERRAL AND REMEDIAL ACTION.

How might we use Matchmaker in practice for schema matching? Let us examine two cases.

**(1) Matchmaker with human-in-the-loop deferral:** We evaluate the effectiveness of integrating Matchmaker with a human-in-the-loop approach by deferring uncertain matches to human experts (i.e., an oracle) for correction. High-uncertainty cases are identified using the entropy of Matchmaker's confidence scores, with the most challenging matches (those with the highest entropy) deferred to the oracle. We evaluate different deferral percentages $p \in \{0, 10, 20, 30, 40, 50\}$ and observe that entropy-based deferral consistently yields greater performance gains compared to random deferral, as shown in Fig. 4(a). This finding highlights the practical value of Matchmaker in real-world settings, where based on entropy, one could strategically seek human oversight for challenging matches and improve overall schema matching performance. The appropriate deferral percentage, however, depends on context-specific factors such as human bandwidth and expert availability.

**(2) Evaluating ease of remedial action based on the similarity between incorrect predictions and true target attributes:** Not all errors in source-target matching are equal; some might be easier to rectify than others. We hypothesize that errors involving semantically similar attributes are easier to correct compared to those involving completely dissimilar attributes. We analyze the cosine similarity between incorrectly predicted attributes and their true target attributes using Pubmed-Bert embeddings. To simulate post-hoc remedial action, we assess the performance gains achieved by correcting erroneous predictions that exceed different similarity thresholds. Figure 4(b) shows substantial improvements in accuracy@1 when "fixing" errors, with high semantic similarity between the erroneous prediction and true attribute (e.g., cosine similarity $\geq 0.8$). These results suggest that Matchmaker's incorrect predictions are often semantically close to the true attributes (i.e. our errors are not far off), making them more amenable to post-hoc remedial actions. This demonstrates the viability of post-hoc remedial actions to improve schema matching performance. Further error analysis can be found in Appendix D.

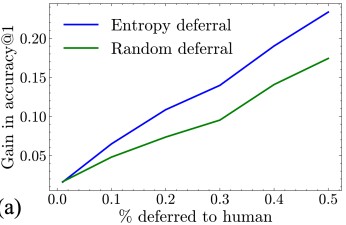
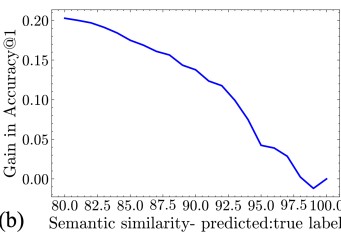

Figure 4: Examples of using Matchmaker in practice. (a) Deferring uncertain samples to humans via entropy deferral improves schema matching performance. (b) Performance gains are obtained when correcting errors which are semantically similar to the true attribute.

## 6 DISCUSSION

Matchmaker introduces a novel approach to schema matching, using a self-improving compositional program using LLMs. Matchmaker's superior performance compared to existing ML-based approaches, underlines its potential to accelerate data integration for ML-ready data. Matchmaker's zero-shot self-improvement mechanism, using synthetic in-context examples, showcases the potential of using LLMs to handle complex reasoning tasks without relying on labeled data.

**Limitations and opportunities.** (1) Matchmaker, while effective in schema matching, represents just one component of the broader data harmonization process and needs to be integrated with other tasks to generate ML-ready data. (2) Despite its advantages over alternative ML-based approaches, Matchmaker is not a panacea and does not achieve perfect automation. It is best used with a human-in-the-loop (Sec. 5.4) to ensure reliability in real-world settings. (3)Future work should address many-many schema matches or explore the viability of non-English schemas.

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

# Appendix - Matchmaker: Schema Matching with self-improving compositional LLM programs

## Table of Contents

## A  MATCHMAKER ADDITIONAL DETAILS

### A.1  MATCHMAKER WITHIN THE CONTEXT OF LLM TABLE REASONING.

There has recently been works on LLMs for table reasoning. We contrast them to Matchmaker along a variety of dimensions below.

**Task/Goal:** The table reasoning papers tackle a variety of tasks centered around understanding and interacting with tabular data. Some examples include: TabSQLify (Nahid & Rafiei, 2024) and OPENTAB (Kong et al., 2023) focus on table question answering and fact verification, aiming to extract relevant information from tables to answer questions or verify statements. Chain-of-Table (Wang et al., 2023b) and "Large Language Models are Few-Shot Table Reasoners" (Chen, 2023) explore LLMs' capabilities in reasoning over tables for question answering and fact verification tasks. The survey paper "Large Language Model for Table Processing" (Lu et al., 2024) covers a broader range of tasks, including table manipulation, table augmentation, and text-to-SQL conversion, showcasing LLMs' potential in interpreting and manipulating tabular data. In contrast, Matchmaker addresses the task of schema matching, which aims to find correspondences between attributes across different schemas or tables. The goal is to enable data integration by mapping attributes from a source schema to a target schema, considering the structural and semantic differences between them. This task is crucial for creating ML-ready datasets by harmonizing data from diverse sources.

**Approach:** Table reasoning approaches span prompting LLMs for direct answers (Chen, 2023), program synthesis to generate SQL/code (Nahid & Rafiei, 2024; Kong et al., 2023), iterative table transformation (Wang et al., 2023b), instruction tuning (Lu et al., 2024), and agent-based methods (Lu et al., 2024). Matchmaker proposes a novel self-improving compositional language model program. It leverages LLM reasoning via a pipeline with multiple LLM calls for candidate generation, refinement and confidence scoring. It also self-improves without labeled data via synthetic in-context examples.

**Inputs:** The table reasoning papers mostly focus on single tables as input along with a question/query. Matchmaker takes as input two tables/schemas (source and target) that need to be matched. It operates solely on schema-level information (attribute names, metadata) without access to raw data in the tables. This is also a key difference compared to the table reasoning papers, which often rely on the actual data values for answering questions or verifying facts.

**Outputs:** Table reasoning papers aim to output answers to questions, binary fact verification labels, updated tables after manipulation, generated SQL/code, etc. In contrast, Matchmaker outputs a mapping between the source and target schema attributes, or indicates no match is possible for certain attributes. The set of attribute pairs representing the schema matching results, can be used to guide data integration processes.

**Use of the LLM:** Table reasoning employs LLMs for direct answer generation (Chen, 2023), program synthesis (Nahid & Rafiei, 2024; Kong et al., 2023), iterative prompting (Wang et al., 2023b), or as part of an agent system (Lu et al., 2024). Matchmaker uses LLMs for reasoning within a compositional program, generating candidates, refining them, and scoring confidence.

**Optimization/Training:** Table reasoning works explore fine-tuning (Nahid & Rafiei, 2024), instruction tuning (Lu et al., 2024), and in-context few-shot learning (Chen, 2023). Matchmaker introduces a novel optimization process to select synthetic in-context examples for self-improvement without labeled data or fine-tuning.

**Key differences:** In summary, while the table reasoning papers focus on tasks like question answering, fact verification, and table manipulation on single tables, Matchmaker addresses the distinct task of schema matching across table pairs. Its novel approach of a self-improving compositional language model program operating on schema-level information contrasts with general table reasoning which mostly use LLMs for direct table QA or program synthesis.

## A.2 MATCHMAKER ALGORITHM

Below we provide a high-level overview algorithm of Matchmakers compositional language model program for schema matching.

---

**Algorithm 2** Matchmaker: Schema Matching with Self-Improving Compositional Language Model Programs

---

**Require:** Source schema $S_s$, Target schema $S_t$
**Ensure:** Schema matches $M$
 1: **Stage 1: Multi-Vector Document Creation**
 2: **for** each table $T \in S_t$ **do**
 3:     Create document $D_T$ with attribute names and descriptions
 4:     Append table metadata to $D_T$
 5:     Encode $D_T$ using ColBERT-v2 to obtain multi-vector representation $V_T$
 6:     Add $V_T$ to vector database $\mathcal{V}$
 7: **end for**
 8: **Stage 2: Candidate Generation**
 9: **for** each source attribute $q_i \in S_s$ **do**
10:     Encode $q_i$ using ColBERT-v2 to obtain query embedding $E_{q_i}$
11:     Retrieve top-k semantic candidates $C_s$ from $\mathcal{V}$ using $E_{q_i}$
12:     Generate reasoning-based candidates $C_R$ using LLM $l_c(q_i, S_t)$
13:     Refine candidate set $C^* \leftarrow l_r(C_s \cup C_R, q_i)$
14: **end for**
15: **Stage 3: Confidence Scoring**
16: **for** each source attribute $q_i \in S_s$ **do**
17:     Format candidate set $C$ as multiple-choice question $Q_i$
18:     **for** each candidate $c_j \in C$ **do**
19:         Compute confidence score $s_j \leftarrow l_s(Q_i, c_j)$
20:     **end for**
21:     $m_i \leftarrow \arg\max_{c_j \in C} s_j$                            ▷ Select match with highest confidence
22:     Add $(q_i, m_i)$ to schema matches $M$
23: **end for**
24: **Self-Improvement Optimization (Over all steps)**
25: Generate evaluation set $D_{eval}$ from unlabeled schemas
26: **for** each example $e_i \in D_{eval}$ **do**
27:     $(\hat{y}_i, \text{trace}_i) \leftarrow \text{Matchmaker}(e_i)$          ▷ Run Matchmaker to get output and traces
28:     $s_i \leftarrow E_l(e_i, \hat{y}i)$                           ▷ Compute evaluation score using LLM $E_l$
29:     Add $(e_i, \text{trace}i, \hat{y}i, s_i)$ to $D_{demo}$
30: **end for**
31: Sort $D_{demo}$ by score $s_i$
32: Select top-n examples from $D_{demo}$ as synthetic in-context examples
33: Update Matchmaker components with selected in-context examples
34: **return** Final output: Schema matches $M$

---

A.3   SCHEMA MATCHING CHALLENGES.

- **Database Heterogeneity**: The number of tables in each schema may differ, i.e., $|T_s| \neq |T_t|$, making it challenging to establish correspondences between attributes across schemas.

- **Structural Heterogeneity**: Schemas may have different architectures, hierarchies, and representational granularity. If we define a hierarchy function $h(T_i)$ that describes the level of nesting within tables, differences in $h(T_{sj})$ and $h(T_{tk})$ for any $j$, $k$ can lead to significant challenges in aligning attributes $A_{sj}$ and $A_{tk}$.

- **Semantic Heterogeneity**: Attributes in different schemas may have the same name but different meanings, or different names but the same meaning. Let $N_i = \{n_{ij} | A_{ij} \in A_i\}$ be the set of attribute names for schema $S_i$. Semantic heterogeneity occurs when $\exists A_{sj} \in A_s, A_{tk} \in A_t : f(A_{sj}) = A_{tk} \wedge n_{sj} \neq n_{tk}$ or when $\exists A_{sj} \in A_s, A_{tk} \in A_t : f(A_{sj}) \neq A_{tk} \wedge n_{sj} = n_{tk}$.

- **Data Type Heterogeneity**: Attributes in different schemas may have different data types, even if they refer to the same concept. Let $d_{ij}$ be the data type of attribute $A_{ij}$. Data type heterogeneity occurs when $\exists A_{sj} \in A_s, A_{tk} \in A_t : f(A_{sj}) = A_{tk} \wedge d_{sj} \neq d_{tk}$.

- **Information Mismatch**: Some attributes in one schema may lack a corresponding match in the other schema. This necessitates reasoning about "no possible match" cases, which is as important as reasoning about possible matches.

- **Unsupervised Nature**: Schema matching is unsupervised, where no labeled data pairs $(A_{sj}, A_{tk})$ are available to train or validate the mappings. This necessitates reliance on the intrinsic structure and semantic information encoded in $A_i$, making the development of an effective mapping function $f$ challenging without external supervision.

## A.4  COMPLEXITY OF THE MIMIC-OMOP TASK

MIMIC-OMOP is a real-world healthcare schema matching task, which is reflective of complex structures, interlinking and hierarchies that can be expected in real-world schema matching tasks. Hence, Matchmakers ability to empirically outperform baselines on these tasks highlights its ability to handle complex schemas.

To illustrate the complexity of the schemas that Matchmaker can handle, Figure 5 illustrates the complex schema structure and multiple tables.

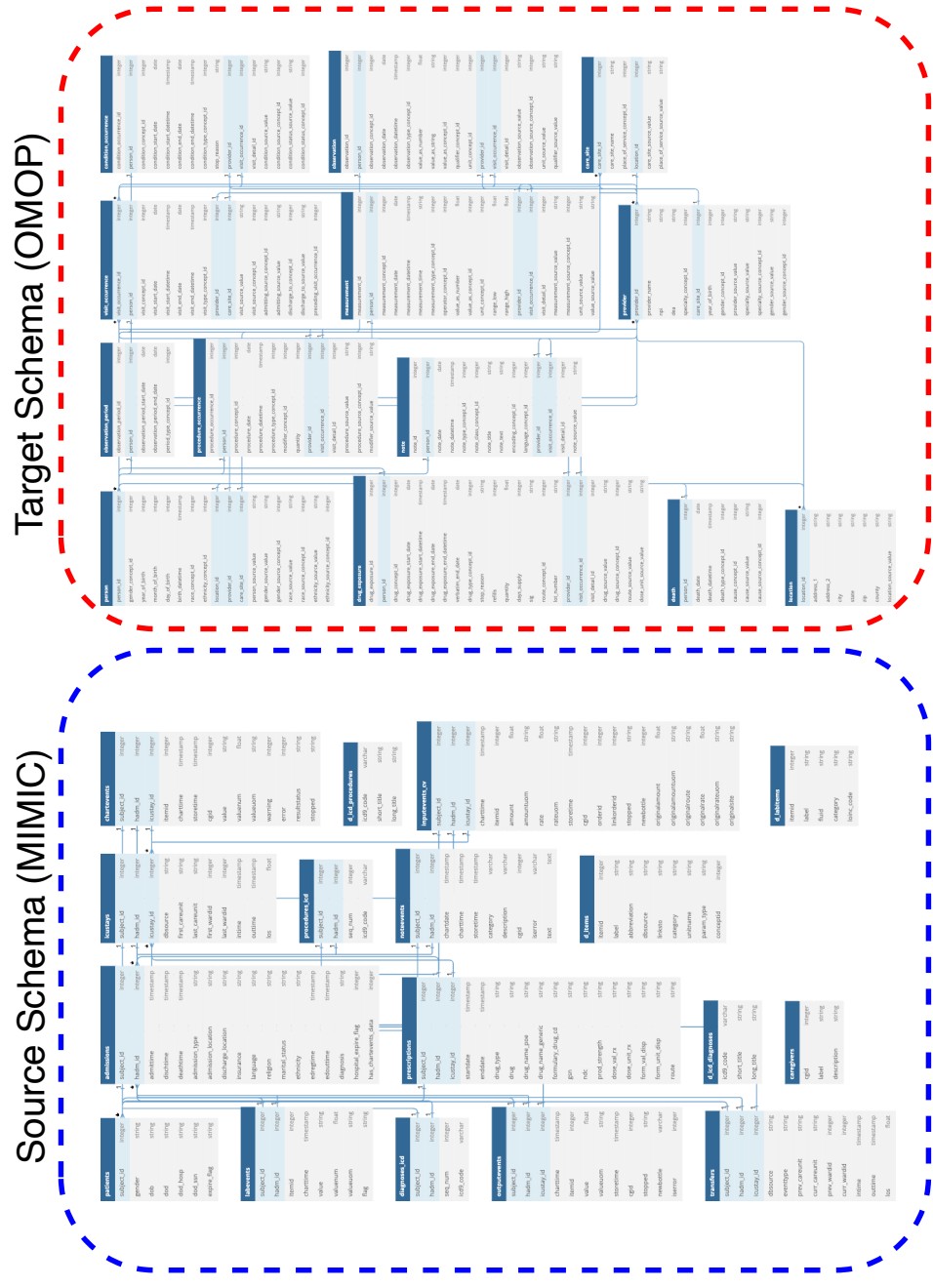

Figure 5: Illustration of the MIMIC-OMOP schema matching task showing the complexity and schema hierarchies.

## A.5 Further details on schema matching formalism

In this appendix, we provide further details on the formulation of schema matching. We look at properties that a schema matching algorithm or function should possess, as well as, detailing how Matchmaker satisfies these properties.

**Properties necessary.** In practice, correctness in schema matching is evaluated against expert-validated ground truth mappings between the datasets (e.g. MIMIC to OMOP and Synthea to OMOP). However, this begs the question what properties would be useful ti improve emprical performance.

These lie along the following dimensions:

- Semantic Equivalence/Consistency: $f(A_S) = A_t$ implies $A$ and $A_t$ represent the same real-world concept (i.e. the mapped attributes serve equivalent purposes)

- Type Compatibility: Mapped attributes must have compatible data types

- Structural Consistency: Mappings must respect schema hierarchies

- Coverage: $f$ identifies all valid matches while avoiding incorrect mappings through abstention. i.e. coverage is maximized by improved accuracy@k

We can then practically assess if a function $f$ (such as Matchmaker) satisfies these criteria based on its performance against expert-validated ground truth mappings in real-world benchmark datasets as has been done in the paper.

**How does Matchmaker satisfy these properties?**

While we have empirically shown Matchmaker satisfies the properties needed of a schema matching function $f$, based on its strong performance on real-world schema matching tasks where it significantly outperforms existing approaches on standard benchmarks. In particular, the strong empirical performance outperforming the baselines implies that Matchmaker better satisfies the properties compared to the baseline schema matching algorithms.

However, let us analyze how Matchmaker also has specific design aspects within its compositional LLM structure that promotes addressing the properties.

- Semantic equivalence/consistency: Matchmaker employs multiple mechanisms: multi-vector document representation captures semantic nuances beyond simple name matching, while dual candidate generation combines both semantic retrieval and LLM reasoning to identify conceptually equivalent attributes.

- Type compatibility: enforced through inclusion of data type information in our multi-vector documents (Section 4.1) and LLM reasoning during candidate generation and refinement (Section 4.2), with examples in Appendix C showing explicit consideration of type compatibility (e.g., string->varchar, integer->bigint).

- Structural consistency is maintained by incorporating table metadata and hierarchical information in document creation (Section 4.1), using reasoning-based candidate generation that considers schema structure (Section 4.2), and including table context in confidence scoring.

- Coverage is optimized through our MCQ format with a "None of the above" option enabling abstention when no good match exists, while confidence scoring helps identify and rank high-quality matches. Our empirical results validate that these properties translate to superior performance in practice.

## A.6 Detailed explanation of self-improvement

The self-improvement mechanism of Matchmaker is a pivotal component. We provide the Algorithm below.

---

**Algorithm 3** Optimize LM program $\mathcal{L}$

---

1: **Input:** Set of evaluation queries $\mathcal{D}_{eval} = e_1, e_2, \ldots, e_n$
2: **Output:** Set of top $n$ demonstrations $D_{demo}$
3: **for** each input $e_i \in \mathcal{D}_{eval}$ **do**
4:     $\hat{y}_i, trace_i \leftarrow \mathcal{L}(e_i)$            $\triangleright$ Teacher $\mathcal{L}$ predicts, storing outputs and intermediate traces
5:     $s_i \leftarrow \mathcal{E}(e_i, \hat{y}_i)$                                 $\triangleright$ Evaluation score
6:     $D_{demo} \leftarrow D_{demo} \cup (e_i, trace_i, \hat{y}_i, s_i)$
7: **end for**
8: Sort $D_{demo}$ by score
9: **return** $D_{demo}[0:n]$                                                $\triangleright$ Select top $n$

---

In particular, we clarify that the self-improvement approach aims to address the issue of in-context learning for multi-stage LLM programs like Matchmaker. However, in doing so we need to address two fundamental challenges in our setting (C1 and C2):

**(C1) Lack of labeled demonstrations**: We do not have access to labeled input-output demonstrations from which to select in-context examples.

**(C2) Lack of an evaluator for selection**: To assess Matchmaker's capabilities and guide selection of examples, we need an evaluator.

We address each as follows:

- Addressing (C1): The process begins by creating an evaluation dataset $D_e val$ from unlabeled schemas with two properties: "easy queries" where top-n semantic matches have similarity scores > 0.95, and "challenging queries" with the lowest semantic match scores. This ensures diverse coverage of different matching scenarios. The complete Matchmaker compositional program L is then run on each evaluation example $e_i \in D_{eval}$. We capture full execution traces including intermediate reasoning steps, candidate generation and refinement decisions, and final confidence scores and matches. The synthetic in-context examples refer to the intermediate input-output pairs generated by the LLM for the intermediate steps of the compositional LLM program. This deals with the challenge of a lack of labeled examples (i.e. zero-shot).

- Addressing (C2): To handle the lack of an evaluator (validation metric), we use an evaluator LLM E (i.e. an LLM-as-a-judge) to assess match quality through chain-of-thought reasoning, producing scores from 0-5 based on match relevance. Finally, the top-n traces are selected based on these evaluation scores. This systematic approach, detailed in Algorithm 1, enables principled selection of in-context examples based on traces that lead to good performance. We then use these as in-context examples for the different parts of the LLM program (as they led to good performance) — in order to guide the reasoning. As shown in the main paper our novel approach to self-improve outperforms random selection of in-context examples and self-reflection confirming that our systematic selection of in-context samples is the key driver of performance gains, rather than the mere inclusion of any in-context examples.

## A.7 EXTENDED RELATED WORK

**Classical Schema Matching approaches.** Classical approaches to schema matching, as thoroughly reviewed by Rahm & Bernstein (2001), use a range of strategies, including heuristic-driven linguistic matching, constraint-based methods, and structural analysis. These methods have historically focused on simple relational schemas, matching elements between individual tables or flat structures. In particular, the primary focus is matching between individual tables or simple schemas (such as purchase orders).

**Key Weaknesses of Classical Approaches and How Matchmaker Addresses Them:**

- Single-Table and Flat Structure Focus: Classical methods typically perform schema matching at the element level, treating tables as isolated entities and matching attributes based on direct comparisons of names, data types, or simple structural cues. In particular, often a focus was simple relational schemas, where the goal was to map elements between single

tables. However, this approach fails to handle the complexity of modern data systems, where schemas are often multi-table, hierarchical, or require cross-table reasoning. **Contrast:** Matchmaker, in contrast, uses LLM-based reasoning to connect attributes across multi-table and hierarchical schemas, understanding how data relationships span multiple tables. This makes our approach significantly more capable of handling complex and interrelated schema structures.

- Dependency on Heuristics and Limited Semantic Understanding: Classical methods rely on heuristic-driven matching based on linguistic similarities (e.g., name matching using synonyms, hypernyms, or edit distance) and structural constraints like key relationships. While these heuristics work in well-defined contexts, they are insufficient for domains where semantic meaning is implicit, such as in healthcare and as we show in Fig 1 — only semantic matching is in fact insufficient. **Contrast:** Matchmaker employs chain-of-thought prompting and advanced LLMs to perform reasoning, allowing it to capture relationships that are not explicitly defined in the schema structure or names. This enables Matchmaker to handle complex mappings that classical methods cannot infer.

- Manual Effort and Lack of Adaptability: Classical techniques require significant manual effort for tuning and adaptation, making them less suitable for rapidly evolving or heterogeneous environments. Constraint-based approaches, in particular, are difficult to scale across different domains without manual intervention. Alternatively, they might also rely on labeled data for effective matching. This makes these classical approaches impractical in real-world environments. **Contrast:** Matchmaker's zero-shot and self-optimization capabilities mean it can adapt autonomously to new schemas using synthetic in-context examples, significantly reducing the need for manual tuning and making it more practical for dynamic, real-world data integration tasks.

**Key Weaknesses of SMAT and how Matchmaker improves:** We also compared Matchmaker to state-of-the-art (SOTA) methods like SMAT Zhang et al. (2021), which applies attention mechanisms for schema matching. While SMAT represents an important advancement over classical methods, it has several limitations that Matchmaker overcomes:

- High Dependency on Labeled Data: SMAT requires extensive labeled data (over 50% labeled matches) for training, which is often impractical in real-world schema matching. **Contrast:** Matchmaker's zero-shot matching capability allows it to perform well without any labeled training data, using LLMs to generate and refine matches autonomously.

- Binary formulation: SMAT formulates the problem as binary classification task over the full Cartesian product of source and target schema attributes. e.g. for each pair of source-target attributes. This leads to a large amount of comparisons. **Contrast:** Matchmakers formulation as information retrieval reduces the number of comparisons and leads to greater efficiency — in addition to the better performance.

A.8 METRICS: ACCURACY, PRECISION, RECALL, F1-SCORE

In our m:1 schema matching setup, accuracy@1, precision, recall, and F1-score are equivalent due to the specific constraints of the task and the prediction mechanism employed. Below, we provide a detailed explanation of this equivalence:

2. **Task Constraints:** The schema matching task is constrained such that each source attribute can match to at most one target attribute (m:1 constraint). This ensures that the number of predictions equals the number of source attributes.

**Equivalence of Metrics** Given the above setup, the following equivalences hold:

**Precision:**
$$\text{Precision} = \frac{\text{True Positives (TP)}}{\text{True Positives (TP)} + \text{False Positives (FP)}}$$

In our setup, every prediction corresponds to exactly one target attribute, and there are no extraneous or unassigned predictions. Therefore:
$$\text{Precision} = \frac{\text{Correct Matches}}{\text{Total Predictions}} = \text{Accuracy@1}.$$

**Recall:**

$$\text{Recall} = \frac{\text{True Positives (TP)}}{\text{True Positives (TP)} + \text{False Negatives (FN)}}$$

Since every source attribute must be matched to a target attribute, there are no unassigned predictions in our setup. However, incorrect matches can occur, leading to both false positives (FP) and false negatives (FN). In our m:1 schema matching setup, a prediction is either correct (a true positive, TP) or incorrect. An incorrect match to the wrong target attribute results in a false positive (FP) for the predicted target and a corresponding false negative (FN) for the true target. Consequently, the number of FP and FN are always equal, as they reflect the same prediction errors. In this setup, precision, recall, and accuracy@1 are equivalent because they all measure the proportion of correct matches (TPs) relative to the total predictions, with incorrect matches impacting all metrics identically. This equivalence holds when correctness is measured against the ground truth annotations from the benchmark datasets. Thus:

$$\text{Recall} = \frac{\text{Correct Matches}}{\text{Total Predictions}} = \text{Accuracy@1}.$$

**F1-Score:**

$$\text{F1-Score} = 2 \cdot \frac{\text{Precision} \cdot \text{Recall}}{\text{Precision} + \text{Recall}}$$

As both precision and recall are equal to accuracy@1 in this setup, the F1-score simplifies to:

$$\text{F1-Score} = \text{Accuracy@1}.$$

In summary, due to the constraints of our m:1 schema matching task and the `argmax` prediction mechanism, accuracy@1, precision, recall, and F1-score are mathematically equivalent. We report accuracy@1 in the main results, but the corresponding precision, recall, and F1-scores are identical and can be directly interpreted from the accuracy@1 values. We note this equivalence does not hold for one-to-many mappings

## B    EXPERIMENTAL DETAILS: BENCHMARKS & DATASETS

All experiments are run on a single Nvidia A4000 GPU with 20 GB of vram. We invoke GPT-4 via the Azure OpenAI API.

### B.1    BENCHMARKS

#### B.1.1    MATCHMAKER

Matchmaker is a compositional language model program for schema matching made up of multiple component modules — formulated in the context of information retrieval.

**GPT-4 Hyper-parameters.** The model version used as the LLM was GPT-4-1106, with the following settings: {'temperature': 0.5, 'max_tokens': 1024, 'top_p': 1, 'frequency_penalty': 0, 'presence_penalty': 0, 'n': 1, }

**Embedding model and documents.** We use Colbert-V2 (Santhanam et al., 2022) as the embedding model and follow the document creation process as outlined in Sec. 4.1. We use the implementation of Colbert-v2 from RAGatouille (https://github.com/bclavie/RAGatouille/).

**Candidates.** For both semantic and reasoning-based candidates, we set k=5.

**Optimization.** As described in the main paper, we generate synthetic in-context samples to address the unique challenges of a lack of labeled data and no demonstrations. As described, to achieve this we follow a boostrapping process like in DSPy (Khattab et al., 2023). For our experiments we select at maximum 4 synthetic in-context examples

**Prompts:** We show examples with the prompts for each component of Matchmaker in Appendix C.

#### B.1.2    REMATCH

In the main text we report the numbers directly from the ReMatch paper, as there is no open-source implementation.

**How we selected the numbers to report:** The ReMatch paper does an exploration of the number of documents retrieved. Hence, we use the following two criteria.
(i) At least 1 document must be retrieved. i.e. the retrieval step cannot be skipped.

(ii) We then select the result that satisfies (i), with the highest accuracy@5.

Our implementation of ReMatch follows the original paper (Sheetrit et al., 2024). We use OpenAI Ada embeddings for the embedding model and GPT-4 as the LLM.

We following the document creation procedure and use the prompt template as provided.

**GPT-4 Hyper-parameters.** The model version used for generation was GPT-4-1106, with the following settings from the ReMatch paper: {seed=42, temperature=0.5, max_tokens=4096, top_p=0.9, frequency_penalty=0, presence_penalty=0}

#### B.1.3    JELLYFISH

Jellyfish (Zhang et al., 2023b) is a fine-tuned language model tailored for data preprocessing tasks including schema matching. The 7B and 13B models are fine tuned upon the OpenOrca-Platypus2 model.

Implementation (7b): https://huggingface.co/NECOUDBFM/Jellyfish-7B

Implementation (13b): https://huggingface.co/NECOUDBFM/Jellyfish-13B

#### B.1.4    LLM-DP

LLM-DP (Narayan et al., 2022; Zhang et al., 2023a) refer to works which have used pre-trained LLMs like GPT-3.5 or GPT-4 for data processing tasks like schema matching via prompting. Since the papers in the few-shot case use labeled examples we do not use those — given its unrealistic in practice. Hence, for these baselines they operate in a zero shot manner.

Implementation: https://github.com/HazyResearch/fm_data_tasks

### B.1.5   SMAT

SMAT is a supervised learning approach which performs schema matching via an attention mecha-
nism. Of course, the model needs labeled data to train on. In our experiments, we assess two variants
given that labeled training data for schema matching is hard to access: (i) 20-80: 20% train and 80%
test and (ii) 50-50: 50% train and 50% test.

We use the default hyper-parameters: {Learning Rate: 0.8, Batch Size: 64, Epochs: 30}

Implementation: https://github.com/JZCS2018/SMAT

### B.2   DATASETS

We outline the two real-world schema matching benchmarks used in this paper — MIMIC and
Synthea. These datasets mapping different clinical/healthcare schemas were chosen as they are
the standard datasets used in schema matching literature and consequently, used by prior works
providing fair assessment. They are also considered the most reflective of real-world schema matching
complexity and challenges. We note that the scarcity of complex and challenging real-world datasets,
underscores the challenges in collecting and annotating real-world schema matching data. For
instance, as noted in Sec 1, annotating MIMIC-OMOP alone required 500 hours from two medical
experts.

Table 5 provides a summary of the table properties.

Note there is no specific train-test sets used as in supervised learning. As we perform the schema
matching task in a zero-shot manner.

Table 5: Summary of the table properties of our two schema matching datasets.

| Dataset | Source Tables | Target Tables |
|---|---|---|
| MIMIC-OMOP | 26 | 14 |
| SYNTHEA-OMOP | 12 | 21 |

**MIMIC Dataset:** The dataset contains a schema mapping between the MIMIC-III electronic health
record (Source schema) (Johnson et al., 2016) and The Observational Medical Outcomes Partnership
Common Data Model (OMOP schema) (Target schema).

This dataset is currently the largest publicly available schema matching dataset (Sheetrit et al., 2024)
and is the cloest to a real-world schema matching use case, wherein a proprietary database created
for a specific purpose (a source schema) is mapped to a given industry standard (a target schema) for
further uses. In this case the proprietary database schema is MIMIC and the industry standard is the
OMOP common data model.

*Open-source data*: https://github.com/meniData1/MIMIC_2_OMOP

**Synthea Dataset:** The Synthea dataset is part of the OMAP benchmark (Zhang et al., 2021) and
is a partial mapping of the Synthea (Walonoski et al., 2018) (Source Schema) which is a synthetic
healthcare dataset of a Massachusetts health records and attempts to map it to a subset of the
OMOP CDM (Target Schema). The dataset has widely been used in previous schema matching
papers (Sheetrit et al., 2024; Narayan et al., 2022; Zhang et al., 2021) as a realistic and challenging
real-world schema matching benchmark.

*Open-source data*: https://github.com/JZCS2018/SMAT/tree/main/datasets/omap/

# C    EXAMPLES USING MATCHMAKER (WITH PROMPTS)

## C.1    MATCHMAKER PROMPT EXAMPLES

We show two end-to-end schema matching examples with Matchmaker, where other methods fail. (1) Example 1: case with No possible target schema match for the source schema query, (2) Example 2: challenging reasoning case, where there is a match possible between source and target schema.
▶ **In each component, we can show the "Optimized" In-context examples.**

### C.1.1    EXAMPLE 1.

**Source schema query:** admissions-marital_status(string): Table admissions details-the admissions table gives information regarding a patient's admission to the hospital., Attribute marital_status details -describe patient demographics.

**Target scheme match:** None possible.

**Matchmaker:** None of the above.

Figure 6: EXAMPLE 1: Candidate generation.

| Candidate generation |
| --- |

You are an OMOP Schema expert. Your goal is to take the OMOP schema and based on the input, refine the schema to include only 5 most likely matches to the input query.
—
Follow the following format.
Input Schema: Input OMOP schema values Input Query: input query Refined Schema: Five most likely matches to input query. Include most likely matches to the input query. Respond with a single JSON object. JSON Schema: {"defs": {"Extractor": {"properties": {"related": {"description": "related matches", "title": "Related", "type": "string"}}, "required": ["related"], "title": "Extractor", "type": "object"}}, "properties": {"value": {"items": {"ref": "/defs/Extractor"}, "title": "Value", "type": "array"}}, "required": ["value"], "title": "Output", "type": "object"}
—
Input Schema: ['visit_occurrence-person_id(bigint)', 'visit_occurrence-visit_occurrence_id(bigint)', ... 'procedure_occurrence-provider_id(bigint)', 'visit_detail-visit_detail_source_value(varchar(50))']
Input Query: procedureevents_mv-itemid
Refined Schema: "value": ["related": "procedure_occurrence-person_id(bigint)", "related": "procedure_occurrence-visit_occurrence_id(bigint)", "related": "procedure_occurrence-procedure_date(date)", "related": "procedure_occurrence-procedure_source_value(varchar(50))", "related": "procedure_occurrence-procedure_concept_id(integer)"]
—
Input Schema: ['visit_occurrence-person_id(bigint)', 'visit_occurrence-visit_occurrence_id(bigint)', ... 'procedure_occurrence-provider_id(bigint)', 'visit_detail-visit_detail_source_value(varchar(50))']
Input Query: noteevents-text
Refined Schema: "value": ["related": "note-note_text(varchar(max))", "related": "note-note_title(varchar(250))", "related": "note-note_source_value(varchar(50))", "related": "note-note_date(date)", "related": "note-note_datetime(datetime)"]
—
Input Schema: ['visit_occurrence-person_id(bigint)', 'visit_occurrence-visit_occurrence_id(bigint)', ... 'procedure_occurrence-provider_id(bigint)', 'visit_detail-visit_detail_source_value(varchar(50))']
Input Query: admissions-marital_status(string): Table admissions details-the admissions table gives information regarding a patient's admission to the hospital., Attribute marital_status details -describe patient demographics.
Refined Schema:
——————————————————————————————

LLM Response
——————————————————————————————-
{"value": [{"related": "person-person_id(bigint)"}, {"related": "person-gender_source_value(varchar(50))"}, {"related": "person-birth_datetime(datetime)"}, {"related": "person-death_datetime(datetime)"}, {"related": "visit_occurrence-admitted_from_source_value(varchar(50))"}]]}

Figure 7: EXAMPLE 1: Candidate refinement.

**Candidate refiner**

You are an expert OMOP matching ranker. Your task is to take the OMOP candidates and based on the input, refine the candidates to select the 5 most likely matches to the input query. Return ONLY the keys.
—
Follow the following format.
Input Schema: List of key: value pairs
Input Query: input query
Reasoning: Let's think step by step in order to {produce the refined_string_list}. We ...
Refined String List: Five most likely matches to input query. Include maximum of the 5 most likely matches to the input query. Return ONLY the keys
—
Input Schema: ['person-person_id(bigint): Table person details-this table serves as the central identity management for all persons in the database. it contains records that uniquely identify each person or patient, and some demographic information., Attribute person_id details -it is assumed that every person with a different unique identifier is in fact a different person and should be treated independently.', ... 'measurement-measurement_concept_id(integer): the measurement_concept_id field is recommended for primary use in analyses, and must be used for network studies']
Input Query: procedureevents_mv-itemid
Reasoning: Let's think step by step in order to produce the refined string list. We are looking for a match to 'procedureevents_mv-itemid', which seems to relate to a procedure event and its item identifier. The 'procedureevents_mv' part suggests we're looking at a procedure occurrence, and the 'itemid' part suggests we're looking for an identifier or id field. Given that, we should look for keys in the 'procedure_occurrence' table that represent identifiers.
Refined String List: 'procedure_occurrence-person_id(bigint)', 'procedure_occurrence-visit_occurrence_id(bigint)', 'procedure_occurrence-procedure_concept_id(integer)', 'procedure_occurrence-visit_detail_id(bigint)', 'procedure_occurrence-provider_id (bigint)'
—
Input Schema: ['person-person_id(bigint): Table person details-this table serves as the central identity management for all persons in the database. it contains records that uniquely identify each person or patient, and some demographic information., Attribute person_id details -it is assumed that every person with a different unique identifier is in fact a different person and should be treated independently.', ... 'measurement-measurement_concept_id(integer): the measurement_concept_id field is recommended for primary use in analyses, and must be used for network studies']
Input Query: noteevents-text
Reasoning: Let's think step by step in order to produce the refined string list. We are looking for matches to "noteevents-text" in the input schema. The term "noteevents" matches with "note" in the input schema, and "text" matches with "note_text". So, we should consider those keys that contain both "note" and "text".
Refined String List: ['note-note_text(varchar(max))', 'note-note_title(varchar(250))', 'note-note_source_value(varchar(50))', 'note-note_text(varchar(max))']
—
Input Schema: ['person-person_id(bigint): Table person details-this table serves as the central identity management for all persons in the database. it contains records that uniquely identify each person or patient, and some demographic information., Attribute person_id details -it is assumed that every person with a different unique identifier is in fact a different person and should be treated independently.', ... 'measurement-measurement_concept_id(integer): the measurement_concept_id field is recommended for primary use in analyses, and must be used for network studies']
Input Query: admissions-marital_status(string): Table admissions details-the admissions table gives information regarding a patient's admission to the hospital., Attribute marital_status details -describe patient demographics.
Reasoning: Let's think step by step in order to
___________________________________________

**LLM Response**
___________________________________________

"produce the refined string list. We are looking for a match to 'admissions-marital_status', which seems to relate to a hospital admission and the marital status of the patient. The 'admissions' part suggests we're looking at a visit occurrence or visit detail, and the 'marital_status' part suggests we're looking for demographic information, which is usually found in the 'person' table. Given that, we should look for keys in the 'visit_occurrence', 'visit_detail', and 'person' tables.String List: 'visit_occurrence-person_id(bigint)', 'visit_occurrence-visit_occurrence_id(bigint)', 'visit_detail-person_id(bigint)', 'visit_detail-visit_occurrence_id(bigint",

Figure 8: EXAMPLE 1: MCQ Formatter.

**MCQ Formatter**

You are an expert MCQ formatter. Your task is to take a list of schema values and convert them into a multiple choice question format with (letter)Schema value, where the schema values should be key(description).
—
Follow the following format.
Input: input list of schema values Mcq: MCQ format of schema values e.g (A)Schema value, (B)Schema value. Do not include additional options, only the schema values as options. where the schema values should be key(description). Add a No Match option.
—
Input: ["'procedure_occurrence-person_id(bigint)', 'procedure_occurrence-visit_occurrence_id(bigint)', 'procedure_occurrence-procedure_concept_id(integer)', 'procedure_occurrence-visit_detail_id(bigint)', 'procedure_occurrence-provider_id (bigint)'"] Mcq: (A)'procedure_occurrence-person_id(bigint)', (B)'procedure_occurrence-visit_occurrence_id(bigint)', (C)'procedure_occurrence-procedure_concept_id(integer)', (D)'procedure_occurrence-visit_detail_id(bigint)', (E)'procedure_occurrence-provider_id (bigint)', (F)No Match
—
Input: ["['note-note_text(varchar(max))', 'note-note_title(varchar(250))', 'note-note_source_value(varchar(50))', 'note-note_text(varchar(max))']"] Mcq: (A)note-note_text(varchar(max)), (B)note-note_title(varchar(250)), (C)note-note_source_value(varchar(50)), (D)note-note_text(varchar(max)), (E)No Match
—
Input: ["'visit_occurrence-person_id(bigint)', 'visit_occurrence-visit_occurrence_id(bigint)', 'visit_detail-person_id(bigint)', 'visit_detail-visit_occurrence_id(bigint"] Mcq:
————————————————————————————
LLM Response
—————————————————————————-
"(A)'visit_occurrence-person_id(bigint)', (B)'visit_occurrence-visit_occurrence_id(bigint)', (C)'visit_detail-person_id(bigint)', (D)'visit_detail-visit_occurrence_id(bigint)', (E)No Match",

Figure 9: EXAMPLE 1: Confidence scoring.

**Confidence scoring**

You are a schema matching expert. Your task is given the input and the MCQ format of the schema, predict the likelihood or relation score from 0-100 of the input query being related to each option. Your scores will be calibrated. If there is no good match score No Match as 100

—

Follow the following format.
Input Mcq: Input MCQ format of schema values
Input Query: input query
Relation: Relation score of input query being related to the option as value. Assess each independently including No Match, returning a score from 0-100 for each. Return with key as MCQ letter e.g (A) and score=value as JSON

—

Input Mcq: (A)'procedure_occurrence-person_id(bigint)', (B)'procedure_occurrence-visit_occurrence_id(bigint)', (C)'procedure_occurrence-procedure_concept_id(integer)', (D)'procedure_occurrence-visit_detail_id(bigint)', (E)'procedure_occurrence-provider_id (bigint)', (F)No Match
Input Query: procedureevents_mv-itemid
Relation: {"(A)": 0, "(B)": 0, "(C)": 0, "(D)": 0, "(E)": 0, "(F)": 100}

—

Input Mcq: (A)note-note_text(varchar(max)), (B)note-note_title(varchar(250)), (C)note-note_source_value(varchar(50)), (D)note-note_text(varchar(max)), (E)No Match
Input Query: noteevents-text
Relation: {"A": 90, "B": 0, "C": 0, "D": 90, "E": 10}

—

Input Mcq: (A)'visit_occurrence-person_id(bigint)', (B)'visit_occurrence-visit_occurrence_id(bigint)', (C)'visit_detail-person_id(bigint)', (D)'visit_detail-visit_occurrence_id(bigint', (E)No Match Input Query: admissions-marital_status(string): Table admissions details-the admissions table gives information regarding a patient's admission to the hospital., Attribute marital_status details -describe patient demographics.
Relation:
―――――――――――――――――――――――

LLM Response
――――――――――――――――――――――-
"{(A): 0, (B): 0, (C): 0, (D): 0, (E): 100}"

### C.1.2 EXAMPLE 2

**Source schema query:** admissions-marital_status(string): Table admissions details-the admissions table gives information regarding a patient's admission to the hospital., Attribute marital_status details -describe patient demographics.

**Target scheme match:** 'procedure_occurrence- quantity

**Matchmaker:** 'procedure_occurrence- quantity

Figure 10: Candidate generation.

---

**EXAMPLE 2: Candidate generation.**

You are an OMOP Schema expert. Your goal is to take the OMOP schema and based on the input, refine the schema to include only 5 most likely matches to the input query.
—
Follow the following format.
Input Schema: Input OMOP schema values Input Query: input query Refined Schema: Five most likely matches to input query. Include maximum of the 10 most likely matches to the input query. Respond with a single JSON object. JSON Schema: {"defs": {"Extractor": {"properties": {"related": {"description": "related matches", "title": "Related", "type": "string"}}, "required": ["related"], "title": "Extractor", "type": "object"}}, "properties": {"value": {"items": {"ref": "/defs/Extractor"}, "title": "Value", "type": "array"}}, "required": ["value"], "title": "Output", "type": "object"}
—
Input Schema: ['visit_occurrence-person_id(bigint)', 'visit_occurrence-visit_occurrence_id(bigint)', .... 'visit_detail-visit_detail_source_value(varchar(50))']
Input Query: procedureevents_mv-itemid Refined Schema: {"value": [{"related": "procedure_occurrence-person_id(bigint)"}, {"related": "procedure_occurrence-visit_occurrence_id(bigint)"}, {"related": "procedure_occurrence-procedure_date(date)"}, {"related": "procedure_occurrence-procedure_source_value(varchar(50))"}, {"related": "procedure_occurrence-procedure_concept_id(integer)"}]}
—
Input Schema: ['visit_occurrence-person_id(bigint)', 'visit_occurrence-visit_occurrence_id(bigint)', .... 'visit_detail-visit_detail_source_value(varchar(50))'] Input Query: noteevents-text Refined Schema: {"value": [{"related": "note-note_text(varchar(max))"}, {"related": "note-note_title(varchar(250))"}, {"related": "note-note_source_value(varchar(50))"}, {"related": "note-note_date(date)"}, {"related": "note-note_datetime(datetime)"}]}
—
Input Schema: ['visit_occurrence-person_id(bigint)', 'visit_occurrence-visit_occurrence_id(bigint)', .... 'visit_detail-visit_detail_source_value(varchar(50))'] Input Query: procedures_icd-seq_num(integer): Table procedures_icd details-contains icd procedures for patients, most notably icd-9 procedures, Attribute seq_num details -provides the order in which the procedures were performed Refined Schema:
"{value: [{related: procedure_occurrence-person_id(bigint)}, {related: procedure_occurrence-visit_occurrence_id(bigint)}, {related: procedure_occurrence-procedure_date(date)}, {related: procedure_occurrence-procedure_concept_id(integer)}, {related: procedure_occurrence-quantity(integer)}]}"

---

Figure 11: EXAMPLE 2: Candidate Refinement.

| Candidate refinement |
| --- |

You are an expert OMOP matching ranker. Your task is to take the OMOP candidates and based on the input, refine the candidates to select the 5 most likely matches to the input query. Return ONLY the keys.

—

Follow the following format.
Input Schema: List of key: value pairs
Input Query: input query
Reasoning: Let's think step by step in order to {produce the refined_string_list}. We ...
Refined String List: Five most likely matches to input query. Include maximum of the 5 most likely matches to the input query. Return ONLY the keys

—

Input Schema: ['procedure_occurrence-person_id(bigint): Table procedure_occurrence details-this table contains records of activities or processes ordered by, or carried out by, a healthcare provider on the patient with a diagnostic or therapeutic purpose., Attribute person_id details -the person_id of the person for whom the procedure is recorded. this may be a system generated code.' .... 'visit_detail-care_site_id(bigint): this field provides information about the care site where the visit detail took place']
Input Query: procedureevents_mv-itemid
Reasoning: Let's think step by step in order to produce the refined string list. We are looking for a match to 'procedureevents_mv-itemid', which seems to relate to a procedure event and its item identifier. The 'procedureevents_mv' part suggests we're looking at a procedure occurrence, and the 'itemid' part suggests we're looking for an identifier or id field. Given that, we should look for keys in the 'procedure_occurrence' table that represent identifiers.
Refined String List: 'procedure_occurrence-person_id(bigint)', 'procedure_occurrence-visit_occurrence_id(bigint)', 'procedure_occurrence-procedure_concept_id(integer)', 'procedure_occurrence-visit_detail_id(bigint)', 'procedure_occurrence-provider_id (bigint)'

—

Input Schema: ['procedure_occurrence-person_id(bigint): Table procedure_occurrence details-this table contains records of activities or processes ordered by, or carried out by, a healthcare provider on the patient with a diagnostic or therapeutic purpose., Attribute person_id details -the person_id of the person for whom the procedure is recorded. this may be a system generated code.' .... 'visit_detail-care_site_id(bigint): this field provides information about the care site where the visit detail took place']
Input Query: noteevents-text
Reasoning: Let's think step by step in order to produce the refined string list. We are looking for matches to "noteevents-text" in the input schema. The term "noteevents" matches with "note" in the input schema, and "text" matches with "note_text". So, we should consider those keys that contain both "note" and "text".
Refined String List: ['note-note_text(varchar(max))', 'note-note_title(varchar(250))', 'note-note_source_value(varchar(50))', 'note-note_text(varchar(max))']

—

Input Schema: ['procedure_occurrence-person_id(bigint): Table procedure_occurrence details-this table contains records of activities or processes ordered by, or carried out by, a healthcare provider on the patient with a diagnostic or therapeutic purpose., Attribute person_id details -the person_id of the person for whom the procedure is recorded. this may be a system generated code.' .... 'visit_detail-care_site_id(bigint): this field provides information about the care site where the visit detail took place']
Input Query: procedures_icd-seq_num(integer): Table procedures_icd details-contains icd procedures for patients, most notably icd-9 procedures, Attribute seq_num details -provides the order in which the procedures were performed
Reasoning: Let's think step by step in order to
"produce the refined string list. We are looking for a match to 'procedures_icd-seq_num', which seems to relate to a procedure occurrence and its sequence number. The 'procedures_icd' part suggests we're looking at a procedure occurrence, and the 'seq_num' part suggests we're looking for a sequential number or order field. Given that, we should look for keys in the 'procedure_occurrence' table that represent order or sequence.
Refined String List: 'procedure_occurrence-person_id(bigint)', 'procedure_occurrence-visit_occurrence_id(bigint)', 'procedure_occurrence-procedure_date(date)', 'procedure_occurrence-procedure_concept_id(integer)', 'procedure_occurrence-quantity(integer)'"

Figure 12: EXAMPLE 2: MCQ Formatter.

---

**MCQ Formatter**

You are an expert MCQ formatter. Your task is to take a list of schema values and convert them into a multiple choice question format with (letter)Schema value, where the schema values should be key(description).

—

Follow the following format.
Input: input list of schema values Mcq: MCQ format of schema values e.g (A)Schema value, (B)Schema value. Do not include additional options, only the schema values as options. where the schema values should be key(description). Add a No Match option.

—

Input: ["'procedure_occurrence-person_id(bigint)', 'procedure_occurrence-visit_occurrence_id(bigint)', 'procedure_occurrence-procedure_concept_id(integer)', 'procedure_occurrence-visit_detail_id(bigint)', 'procedure_occurrence-provider_id (bigint)'"] Mcq: (A)'procedure_occurrence-person_id(bigint)', (B)'procedure_occurrence-visit_occurrence_id(bigint)', (C)'procedure_occurrence-procedure_concept_id(integer)', (D)'procedure_occurrence-visit_detail_id(bigint)', (E)'procedure_occurrence-provider_id (bigint)', (F)No Match

—

Input: ["['note-note_text(varchar(max))', 'note-note_title(varchar(250))', 'note-note_source_value(varchar(50))', 'note-note_text(varchar(max))']"] Mcq: (A)note-note_text(varchar(max)), (B)note-note_title(varchar(250)), (C)note-note_source_value(varchar(50)), (D)note-note_text(varchar(max)), (E)No Match

—

Input: ["'procedure_occurrence-person_id(bigint)', 'procedure_occurrence-visit_occurrence_id(bigint)', 'procedure_occurrence-procedure_date(date)', 'procedure_occurrence-procedure_concept_id(integer)', 'procedure_occurrence-quantity(integer)'"] Mcq:
"(A)'procedure_occurrence-person_id(bigint)', (B)'procedure_occurrence-visit_occurrence_id(bigint)', (C)'procedure_occurrence-procedure_date(date)', (D)'procedure_occurrence-procedure_concept_id(integer)', (E)'procedure_occurrence-quantity(integer)', (F)No Match",

---

Figure 13: EXAMPLE 2: Confidence scoring.

---

**Confidence scoring**

You are a schema matching expert. Your task is given the input and the MCQ format of the schema, predict the likelihood or relation score from 0-100 of the input query being related to each option. Your scores will be calibrated. If there is no good match score No Match as 100

—

Follow the following format.
Input Mcq: Input MCQ format of schema values Input Query: input query Relation: Relation score of input query being related to the option as value. Assess each independently including No Match, returning a score from 0-100 for each. Return with key as MCQ letter e.g (A) and score=value as JSON

—

Input Mcq: (A)'procedure_occurrence-person_id(bigint)', (B)'procedure_occurrence-visit_occurrence_id(bigint)', (C)'procedure_occurrence-procedure_concept_id(integer)', (D)'procedure_occurrence-visit_detail_id(bigint)', (E)'procedure_occurrence-provider_id (bigint)', (F)No Match Input Query: procedureevents_mv-itemid Relation: {"(A)": 0, "(B)": 0, "(C)": 0, "(D)": 0, "(E)": 0, "(F)": 100}

—

Input Mcq: (A)note-note_text(varchar(max)), (B)note-note_title(varchar(250)), (C)note-note_source_value(varchar(50)), (D)note-note_text(varchar(max)), (E)No Match Input Query: noteevents-text Relation: {"A": 90, "B": 0, "C": 0, "D": 90, "E": 10}

—

Input Mcq: (A)'procedure_occurrence-person_id(bigint)', (B)'procedure_occurrence-visit_occurrence_id(bigint)', (C)'procedure_occurrence-procedure_date(date)', (D)'procedure_occurrence-procedure_concept_id(integer)', (E)'procedure_occurrence-quantity(integer)', (F)No Match Input Query: procedures_icd-seq_num(integer): Table procedures_icd details-contains icd procedures for patients, most notably icd-9 procedures, Attribute seq_num details -provides the order in which the procedures were performed Relation: "{(A): 0, (B): 0, (C): 0, (D): 0, (E): 90, (F): 10}",

## C.2 LLM EVALUATOR

We provide examples of the LLM evaluator, showing demonstrations achieving high and low scores.

---

**LLM Evaluator Example: High Score**

You are a schema matching expert, your task is to rate if any of the suggested matches are potential good matches for the query. Be lenient and rate a match as good (4 or 5) if it is relevant to the query. Rate the matches from 1-5. If none of the matches are good, rate 0.

—

Follow the following format.
Query: The query.
Answers: possible matches
Reasoning: Let's think step by step in order to {produce the rating}. We ...
Rating: Rate if any of the suggested matches are good for the query from 1-5. Only output the rating and nothing else.

—

Query: noteevents-chartdate
Answers: ["'note-note_date(date)', 'note-note_datetime(datetime)',
'note-visit_occurrence_id(bigint)', 'note-provider_id(bigint)', 'note-person_id(bigint)'"]
Reasoning: Let's think step by step in order to

———————————————————————————————

LLM Response

———————————————————————————————-

produce the rating. We can see that the query is looking for a match related to the date of a note event in a chart. The first two suggested matches, 'note-note_date(date)' and 'note-note_datetime(datetime)', are both related to dates and notes, making them potentially good matches for the query. The other three suggestions are not directly related to dates or note events, so they are not as good matches.
Rating: 4

Figure 14: LLM evaluator example, rated with a high score.

**LLM Evaluator Example: Low Score**

You are a schema matching expert, your task is to rate if any of the suggested matches are potential good matches for the query. Be lenient and rate a match as good (4 or 5) if it is relevant to the query. Rate the matches from 1-5. If none of the matches are good, rate 0.
—
Follow the following format.
Query: The query.
Answers: possible matches
Reasoning: Let's think step by step in order to {produce the rating}. We ...
Rating: Rate if any of the suggested matches are good for the query from 1-5. Only output the rating and nothing else.

—
Query: cptevents-subsectionheader
Answers: ['visit_occurrence-visit_type_concept_id(int)',
'visit_detail-visit_detail_concept_id(int)', 'procedure_occurrence-procedure_concept_id(int)',
'procedure_occurrence-visit_detail_id(int)', 'condition_occurrence-visit_occurrence_id(int)']
Reasoning: Let's think step by step in order to
_______________________________________________

LLM Response
_______________________________________________

"produce the rating. We can see that the query is looking for a match related to 'cptevents-subsectionheader'. However, none of the suggested matches seem to directly relate to this query. They all refer to different concepts such as 'visit_occurrence', 'visit_detail', 'procedure_occurrence', and 'condition_occurrence'. Therefore, none of the matches seem to be a good fit for the query.
Rating: 0

Figure 15: LLM evaluator example, rated with a low score.

# D  ADDITIONAL EXPERIMENTS

## D.1  NUMBER OF LLM CALLS

**Goal.** To compare the number of LLM calls required by Matchmaker and other baseline methods for schema matching on the MIMIC-OMOP and SYNTHEA-OMOP datasets.

**Experiment.** We count the number of LLM calls made by each method during the schema matching process on both the MIMIC-OMOP and SYNTHEA-OMOP datasets. For methods that do not rely on LLMs (e.g., SMAT), we consider the number of forward passes through the neural network as equivalent to an LLM call for comparison purposes.

**Results.** Table 6 presents the number of LLM calls required by each method on the two datasets.

Table 6: Number of LLM calls

| Method | MIMIC-OMOP | SYNTHEA-OMOP |
|---|---|---|
| Matchmaker | 1340 | 890 |
| ReMatch | 268 | 178 |
| Jellyfish-13b | 24771 | 29637 |
| Jellyfish-7b | 24771 | 29637 |
| LLM-DP | 24771 | 29637 |
| SMAT | 24771 | 29637 |

**Discussion.** The results in Table 6 highlight the efficiency of Matchmaker and ReMatch in terms of the number of LLM calls required for schema matching.

Both Matchmaker and ReMatch formulate schema matching as an information retrieval problem, which significantly reduces the search space compared to the binary classification formulation used by Jellyfish-13b, Jellyfish-7b, LLM-DP, and SMAT.

The high number of LLM calls required by Jellyfish-13b, Jellyfish-7b, LLM-DP, and SMAT can be attributed to their formulation of schema matching as a binary classification problem over the Cartesian product of source and target attributes. In this formulation, the LLM is prompted to provide a label of Yes/No for each pair of source-target attributes, resulting in a large number of LLM calls that scales ($O(n^2)$). Consequently, these methods are computationally expensive and less scalable compared to Matchmaker and ReMatch, which employ a more efficient approach.

The fewer number of LLM calls used by Matchmaker and ReMatch has practical implications in terms of computational cost and runtime efficiency. By reducing the number of LLM calls, these methods can perform schema matching more quickly and with lower computational overhead compared to methods that rely on a large number of calls. This is particularly important when dealing with large-scale schemas or when schema matching needs to be performed frequently in real-world applications.

## D.2  MATCHMAKER WITH OTHER LLMS

**Goal.** To understand the performance of Matchmaker when using a less powerful LLM backbone compared to GPT-4, and contrast it with the ReMatch baseline using GPT-4.

**Experiment.** We evaluate the performance of Matchmaker using GPT-3.5 as the backbone LLM for all components, instead of GPT-4 which was used in the main experiments. We compare this to the performance of Matchmaker with GPT-4 and ReMatch with GPT-4. All other aspects of the setup remain the same as in the main text.

**Results.** Table 7 shows the schema matching accuracy@k for the different methods. We observe that Matchmaker with GPT-3.5 performs worse than Matchmaker with GPT-4, which is expected given GPT-3.5 is a less powerful LLM. Interestingly, Matchmaker with GPT-3.5 achieves comparable performance to ReMatch with GPT-4, despite GPT-3.5 being a much weaker LLM than GPT-4. On MIMIC, Matchmaker with GPT-3.5 slightly outperforms ReMatch with GPT-4 for accuracy@1 and is competitive at higher k. On Synthea, performance is similar for accuracy@1 but Matchmaker with GPT-3.5 outperforms ReMatch with GPT-4 for accuracy@3 and accuracy@5.

Table 7: Comparison of schema matching performance of different baselines.

| | | Matchmaker (GPT-4) | Matchmaker (GPT-3.5) | ReMatch (GPT-4) |
|---|---|---|---|---|
| MIMIC | acc@1 | **62.20 ± 2.40** ↑ | 48.30± 2.80 ↑ | 42.50 |
| | acc@3 | **68.80 ± 2.00** | 62.00 ± 4.20 | 63.80 |
| | acc@5 | **71.10 ± 2.00** | 70.00 ± 4.20 | **72.90** |
| Synthea | acc@1 | **70.20 ± 1.70** | 47.80 ± 3.20 | 50.50 |
| | acc@3 | **78.60 ± 2.50** | 63.30 ± 4.30 ↑ | 58.10 |
| | acc@5 | **80.90 ± 1.10** | 77.10 ± 0.70 ↑ | 74.30 |

**Discussion.** These results demonstrate that the Matchmaker approach of using a compositional LLM program is quite robust and can provide good schema matching performance even with weaker LLM backbones. The fact that Matchmaker with GPT-3.5 is competitive with ReMatch using GPT-4 highlights the strength of the multi-stage Matchmaker approach over ReMatch's single-stage LLM usage. However, using a more powerful LLM like GPT-4 still provides significant gains, underlining the importance of using an LLM with powerful reasoning capabilities for this complex task.

### D.3 FURTHER PERFORMANCE RESULTS: REMATCH REIMPLEMENTATION

**Goal.** To compare the performance of Matchmaker against the ReMatch baseline, using both the original reported results from the ReMatch paper and the re-implementation of ReMatch.

**Experiment.** In the main paper, we report the performance of the ReMatch baseline using the results directly from the paper, as code is not available for ReMatch. However, for completeness, we also re-implement the ReMatch approach based on the details provided in the ReMatch paper.

Our re-implementation uses the OpenAI Ada-002 text embeddings for the retrieval step, following the same procedure as ReMatch for chunking and creating documents. We then use the same prompts as described in the ReMatch paper for the schema matching task. We compare the performance of our re-implemented ReMatch with the original reported results and Matchmaker.

**Results.** Table 8 presents the schema matching accuracy@k for Matchmaker, the original ReMatch results, and our re-implemented ReMatch. We observe that Matchmaker consistently outperforms both the original ReMatch results and our re-implementation across all metrics and datasets. We also find the re-implemented ReMatch achieves lower performance compared to the original reported results.

Table 8: Comparison of schema matching performance of different baselines.

| | | Matchmaker | ReMatch (Original) | ReMatch (Reimplemented) |
|---|---|---|---|---|
| MIMIC | acc@1 | **62.20 ± 2.40** | 42.50 | 41.99 ± 0.61 |
| | acc@3 | **68.80 ± 2.00** | 63.80 | 46.63 ± 1.99 |
| | acc@5 | **71.10 ± 2.00** | **72.90** | 46.63 ± 1.99 |
| Synthea | acc@1 | **70.20 ± 1.70** | 50.50 | 29.10 ± 0.80 |
| | acc@3 | **78.60 ± 2.50** | 58.10 | 32.71 ± 0.35 |
| | acc@5 | **80.90 ± 1.10** | 74.30 | 33.46 ± 0.63 |

**Discussion.** These results further confirm the superiority of Matchmaker over the ReMatch baseline, even when considering our re-implementation of the method. The lower performance of the re-implemented ReMatch compared to the original reported results could be due to differences in implementation details, such as the choice of text embeddings or variations not accounted for. However, it is important to note that even with these differences, Matchmaker consistently outperforms ReMatch (original) by a significant margin. The fact that Matchmaker achieves strong performance gains over both the original ReMatch and our re-implementation underscores the value of the novel techniques introduced in Matchmaker, such as the multi-stage language model program, the use of diverse candidate generators and the self-improvement mechanism through synthetic in-context examples.

### D.4 IMPROVING PERFORMANCE: USE OF EXISTING MAPPINGS TO REMEDY ERRORS

**Goal.** To investigate the potential performance improvement in Matchmaker when leveraging readily available mappings to rectify errors between directly mapped attributes.

**Experiment.** In schema matching, certain attributes like source_value and concept_id have a direct mapping (e.g. in OMOP). If Matchmaker incorrectly maps the source attribute to the wrong target

attribute (e.g., mapping to source_value instead of concept_id or vice versa), these errors can be easily rectified by leveraging the existing relationship.

To simulate this error correction, we implement a post-processing step where we adjust Matchmaker's predictions if the predicted target attribute has a direct mapping to the true target attribute. We apply this correction for all values of k and measure the resulting performance improvement.

**Results.** Figure 16 shows the accuracy gains across different values of k when applying the mapping correction. We observe consistent performance improvements across all k values. These results indicate that leveraging knowledge can indeed help rectify some of the errors made by Matchmaker.

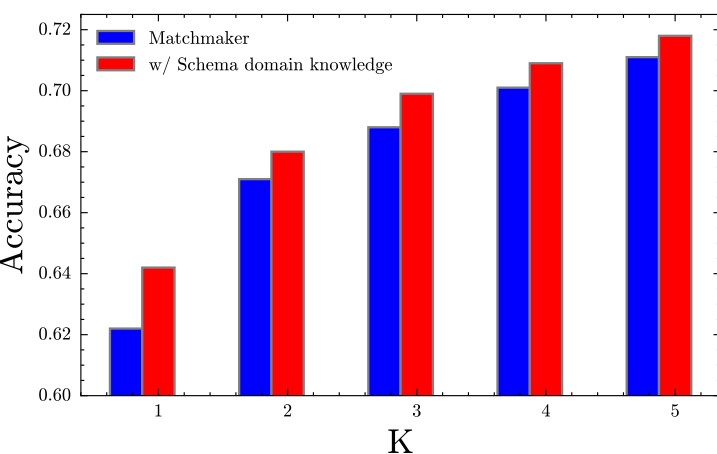

Figure 16: Performance improvement in Matchmaker when leveraging readily available mappings to correct errors between directly mapped attributes like source_value and concept_id.

**Discussion.** While the results demonstrate the potential benefit of using existing mappings for error correction, it is important to note that the performance gains are relatively modest compared to other strategies like human-in-the-loop deferral based on Matchmaker's confidence scores (as shown in the main text).

Moreover, the mapping correction relies on the availability of direct mappings between attributes, which may not always exist in practice. Therefore, while this approach can serve as a useful post-processing step, it should be seen as a complementary technique to be used alongside other strategies like human-in-the-loop for improving schema matching performance.

### D.5 COMPARISON OF MATCHMAKER ON ONTOLOGY MATCHING TASKS

While Schema matching and ontology matching are seemingly related, in reality they are completely different tasks. Specifically, schema and ontology matching fundamentally differ in their task and available information. Ontology matching leverages richer contextual info, including properties, axioms, rules, concept hierarchies and additional annotations. In contrast, schemas are sparser, with only attribute names, data types, descriptions and links.

Despite the difference for completeness we evaluate recent LLM ontology match methods using GPT-4 backbones to mirror Matchmaker namely: OLaLa (Hertling & Paulheim, 2023) and LLMs4OM (Giglou et al., 2024).

As shown in Table 9, Matchmaker outperforms these methods on both datasets.

Table 9: Accuracy@1: Matchmaker vs two LLM-based Ontology matching methods.

| Method | MIMIC | Synthea |
|---|---|---|
| Olala | $33.58 \pm 0.47$ | $31.53 \pm 3.37$ |
| LLMs4OM | $44.78 \pm 0.41$ | $64.50 \pm 2.02$ |
| Matchmaker (Ours) | $\mathbf{62.20 \pm 2.40}$ | $\mathbf{70.20 \pm 1.70}$ |

### D.6 Detailed error analysis

**Goal.** We wish to understand different dimensions of Matchmaker's errors.

**Discussion.** We analyze the errors made by Matchmaker and find two categories of errors.

- 17% of Matchmaker's errors occur when attempting to find matches for source attributes that have no corresponding target attribute.
- The remaining 83% involve selecting incorrect but semantically related attributes. For these incorrect matches, we find a mean semantic similarity of 0.862 between the erroneous predicted attribute and the true target attribute. This confirms that Matchmaker typically selects attributes semantically close to the correct match rather than completely unrelated attributes.

These results further provide an understanding of Matchmaker's errors, as well as, showing how they can be addressed both via uncertainty deferral and remediation being easy to identify and correct.

### D.7 Ranking ablation

**Goal.** Assess the importance of ranking to Matchmakers performance.

**Results.** Below we ablate the ranking step. The results shown highlight the importance of the re-ranking step towards achieving better accuracy@1.

Table 10: Comparison of Matchmaker models with and without ranking on MIMIC and Synthea datasets.

|  |  | Matchmaker (with ranking) | Matchmaker (No ranking) |
|---|---|---|---|
| **MIMIC** | Acc@1 | 62.20 | 57.00 |
|  | Acc@3 | 68.80 | 66.90 |
|  | Acc@5 | 71.10 | 71.10 |
| **Synthea** | Acc@1 | 70.20 | 62.40 |
|  | Acc@3 | 78.60 | 77.20 |
|  | Acc@5 | 80.90 | 80.90 |

# E  BROADER IMPACT

Schema matching is a critical step in data integration, enabling the creation of large, harmonized datasets that can be used to train machine learning models. The proposed Matchmaker approach, with its self-improving compositional language model program, has the potential to significantly accelerate and automate the schema matching process, thus facilitating the development of more accurate and robust ML models across various domains.

The importance and value of schema matching cannot be overstated, as integrating data from various sources such as different regions, organizations or applications is vital in many fields, including healthcare, finance, and e-commerce. By enabling the integration of data from disparate sources, schema matching plays a critical role in creating comprehensive, harmonized datasets that can provide a more complete picture of the domain under study. For example, in healthcare, integrating data from multiple hospitals can lead to more representative and diverse datasets, allowing researchers to identify patterns and insights that may not be apparent when analyzing data from a single institution.

Moreover, schema matching is not only valuable for specific domains but also for the machine learning community as a whole. By increasing the pool of available data (larger and more diverse) for training and validation, schema matching can contribute to the development of more accurate, robust, and generalizable ML models. Furthermore, having access to a larger pool of data can enable more rigorous validation and testing of ML models, allowing researchers to assess their performance across different subpopulations, time periods, and data sources. This, in turn, can lead to more reliable and trustworthy ML models that can be confidently applied in real-world settings.

Below we describe some positive implications that could be unlocked as schema matching approaches such as Matchmaker are used in practice. We also show some drawbacks with mitigation strategies.

**Positive Impacts:**

- Improved data integration: Matchmaker can help overcome the challenges of integrating data from heterogeneous sources, leading to the creation of larger, more comprehensive datasets. This can enable the development of more powerful and generalizable ML models.

- Accelerated research and discovery: By reducing the time and effort required for data integration, Matchmaker can accelerate research and discovery in fields, where data often resides in disparate databases with diverse schemas.

- Enhanced decision-making: The ability to train ML models on larger, more diverse datasets enabled by Matchmaker can lead to more accurate and reliable predictions, supporting better decision-making in various applications.

**Potential Drawbacks and Mitigation Strategies:**

- Overreliance on automated schema matching: While Matchmaker can significantly automate the schema matching process, it is not perfect and may make errors. Overreliance on automated methods without human oversight could lead to incorrect data integration. Mitigation: Matchmaker should be used as a tool to assist human experts in the schema matching process, rather than as a complete replacement. The paper demonstrates how Matchmaker can be effectively used with a human-in-the-loop approach, leveraging the strengths of both human expertise and automated methods.

- Propagation of errors: If Matchmaker introduces errors during the schema matching process, these errors can propagate downstream and affect the quality of the resulting integrated datasets and ML models. Mitigation: It is essential to implement rigorous validation and quality control measures to detect and correct errors introduced by Matchmaker. This can include manual spot-checks, automated consistency checks, and the use of domain-specific validation rules. Establishing a feedback loop to continuously monitor and improve Matchmaker's performance based on real-world usage can also help mitigate the propagation of errors.

