# OpenReview forum: "Matchmaker: Schema Matching with self-improving compositional LLM programs"
_ICLR.cc/2025/Conference — Submitted to ICLR 2025_

### Official Review · Reviewer_JXrD · 2024-10-30

**Soundness:** 4
**Presentation:** 4
**Contribution:** 3
**Rating:** 8
**Confidence:** 4

**Summary:**

This paper introduces Matchmaker, a novel approach to schema matching using a self-improving compositional language model program. Schema matching is crucial for creating interoperable ML-ready data by finding correspondences between attributes across different databases. Matchmaker operates through three main components: multi-vector document creation, candidate generation (using both semantic retrieval and LLM-based reasoning), and confidence scoring. A key innovation is its ability to self-improve without labeled data through synthetic in-context examples. The method significantly outperforms existing approaches on real-world healthcare schema matching benchmarks (MIMIC-OMOP and Synthea-OMOP).

**Strengths:**

1. Addresses a critical practical problem in data integration that has significant implications for ML development
2. Novel technical approach combining retrieval and LLM reasoning in a compositional program
3. Zero-shot learning capability through synthetic in-context examples, eliminating the need for labeled training data
4. Comprehensive empirical evaluation against multiple baselines
5. Practical considerations for deployment, including human-in-the-loop integration and uncertainty handling
6. Strong quantitative results showing 15-20% improvement over baselines
7. Well-documented implementation details and ablation studies

**Weaknesses:**

1. Limited evaluation to two healthcare domain datasets, though the approach is claimed to be general
2. The synthetic in-context example generation process could be explained more clearly
3. The paper shows strong performance metrics but doesn't provide detailed analysis of where and why Matchmaker fails

**Questions:**

1. How does the performance of Matchmaker scale with very large schemas (>1000 attributes)?
2. Could the approach be extended to handle many-to-many mappings between schemas?
3. How sensitive is the performance to the quality of attribute descriptions in the schemas?
4. What strategies could be employed to reduce the number of LLM calls while maintaining performance?
5. How might the system handle schemas in languages other than English?

---

> ### Author Response · Authors · 2024-11-18
> **Response to Reviewer JXrD [Part 1/3]**
>
> Dear ``Reviewer JXrD``
>
> Thank you for your thoughtful comments and suggestions! We give answers to each of the following in turn, along with corresponding updates to the revised manuscript.
>
> (A) Clarifying our use of benchmark evaluation datasets [**PART 1/3**]
>
> (B) Clarifying synthetic in-context examples [**PART 1/3**]
>
> (C) Additional analysis of failure cases [**PART 2/3**]
>
> (D) Questions [**PART 3/3**]
>
>
>
> ----
>
> ## **(A) Clarifying our use of benchmark evaluation datasets**
>
> Our evaluation focused on the MIMIC and Synthea datasets for the following reasons.
>
> (1)   **Standard benchmark datasets**: these datasets are the standard benchmark datasets used by all our baseline schema matching works. In fact, sometimes Synthea has ONLY been used (e.g. LLM-DP).
>
> (2)   **Reflect real-world complexity**:These datasets are used as the benchmarks in the literature as they reflect the complexity of real-world schema matching and are widely accepted in the field.  The scarcity of similar datasets underscores the challenges in collecting and annotating real-world schema matching data. For instance, as noted in Sec 1, annotating MIMIC-OMOP alone required 500 hours from two medical experts.
>
> (3)   **Exhibit major challenges**: They exhibit all major schema matching challenges outlined in Section 3.2: database heterogeneity (different numbers of tables), structural heterogeneity (varied hierarchies), semantic heterogeneity (same concepts with different names), and information mismatch.
>
> Matchmaker's strong performance on these challenging datasets demonstrates its ability to handle the fundamental challenges of schema matching, independent of domain. I.e. While our initial evaluation is domain-specific, the architecture of Matchmaker is domain-agnostic. The key components—semantic retrieval, LLM-based reasoning, and self-improvement via synthetic examples—are applicable to any schema matching task where semantic ambiguity and structural heterogeneity exist.
>
>
>
>
> ----
>
> ## **(B) Clarifying synthetic in-context examples**
>
> The synthetic in-context examples is a vital component for Matchmaker's self-improvement, and we understand the need for clarification. In particular, we clarify that the self-improvement approach aims to address the issue of in-context learning for multi-stage LLM programs like Matchmaker. However, in doing so we need to address two fundamental challenges in our setting (C1 and C2):
>
> **(C1) Lack of labeled demonstrations:** We do not have access to labeled input-output demonstrations from which to select in-context examples.
>
> **(C2) Lack of an evaluator for selection:** To assess Matchmaker’s capabilities and guide the selection of examples, we need an evaluator.
>
> Below we include a more detailed explanation of **Algorithm 1**:
>
> **Addressing (C1):** The process begins by creating an evaluation dataset D_eval from unlabeled schemas with two properties: "easy queries" where top-n semantic matches have similarity scores > 0.95, and "challenging queries" with the lowest semantic match scores. This ensures diverse coverage of different matching scenarios. The complete Matchmaker compositional program L is then run on each evaluation example $e_i \in D_eval$. We capture full execution traces including intermediate reasoning steps, candidate generation and refinement decisions, and final confidence scores and matches. The synthetic in-context examples refer to the intermediate input-output pairs generated by the LLM for the intermediate steps of the compositional LLM program. This deals with the challenge of a lack of labeled examples (i.e. zero-shot)
>
> **Addressing (C2):** To handle the lack of an evaluator (validation metric), we use an evaluator LLM E (i.e. an LLM-as-a-judge) to assess match quality through chain-of-thought reasoning, producing scores from 0-5 based on match relevance. Finally, the top-n traces are selected based on these evaluation scores. This systematic approach, detailed in Algorithm 1, enables principled selection of in-context examples based on traces that lead to good performance. We then use these as in-context examples for the different parts of the LLM program (as they led to good performance) --- to guide the reasoning. As shown in Table 3 our novel approach to self-improve outperforms random selection of in-context examples and self-reflection confirming that our systematic selection of in-context samples is the key driver of performance gains, rather than the mere inclusion of any in-context examples.
>
> **UPDATE:** We provide this more detailed explanation of Algorithm 1 in a new Appendix A.6

---

> ### Author Response · Authors · 2024-11-18
> **Response to Reviewer JXrD [Part 2/3]**
>
> ## **(C) Additional analysis on failure cases**
>
> We clarify we do carry out an error analysis to understand where Matchmaker makes errors and the feasibility of remedial action. In **Section 5.4**, we demonstrate that Matchmaker’s prediction errors are often semantically similar to the true target attributes, as shown in Figure 4(b), where correcting matches with high semantic similarity (≥ 0.8) leads to substantial improvements in accuracy@1. The paper also introduces practical strategies for identifying potential errors through uncertainty quantification, with Figure 4(a) showing that entropy-based deferral to human experts yields greater performance gains compared to random deferral.
>
> **New experiment:** Based on your comment,  we have conducted a new experiment to further analyze the errors: 17% of Matchmaker's errors occur when attempting to find matches for source attributes that have no corresponding target attribute, while the remaining 83% involve selecting incorrect but semantically related attributes. For these incorrect matches, we find a mean semantic similarity of 0.862 between the erroneous predicted attribute and the true target attribute.
>
> This confirms that Matchmaker typically selects attributes semantically close to the correct match rather than completely unrelated attributes. These results further provide an understanding of Matchmaker’s errors, as well as, showing how they can be addressed both via uncertainty deferral and remediation being easy to identify and correct.
>
>  **UPDATE:** Added a new Appendix D.6 with the detailed error analysis

---

> ### Author Response · Authors · 2024-11-18
> **Response to Reviewer JXrD [Part 3/3]**
>
> ## **(D) Questions**
>
> - ``Q1: How Matchmaker scales to large schemas``
>
> Regarding scaling performance for >1000 attributes, our information retrieval formulation provides scalability advantages over traditional binary classification approaches (which are O(n^2)). As shown in Table 6 (Appendix D.1), Matchmaker requires significantly fewer LLM calls compared to most baselines (1340 vs 24771 for MIMIC-OMOP). Specifically, our method employs semantic retrieval and reasoning-based candidate generation to limit the number of attribute comparisons, resulting in a more scalable and efficient process compared to traditional methods. Of course, we add more LLM calls compared to our other benchmark Re-Match but this cost results in superior performance.
>
> Hence, for greater than 1000 attributes while the number of comparisons in Matchmaker naturally increases with the number of attributes, our design ensures a more efficient scaling behavior than binary classification approaches, making it practical for large schemas.
>
>
> - ``Q2: Handling Many-to-Many Mappings``
>
> Our current implementation focuses on many-to-one or one-to-one mappings, which are the most common in practice — evidenced by our healthcare benchmarks and the related work. However, we thank the reviewer for bringing this up as we believe future work could extend ideas from Matchmaker to support many-to-many mappings by modifying the candidate generation. Additionally, the MCQ confidence scoring mechanism could be adapted to allow multiple valid matches rather than scoring a single Match, while our refinement step could also consider multiple possible matches. This represents an exciting direction for future work that builds naturally on our current architecture.
>
> **UPDATE:** We have updated the discussion to include this.
>
> - ``Q3: Matchmaker’s sensitivity to attribute descriptions``
>
> Attribute names being descriptive/informative is indeed a critical factor for Matchmaker. We clarify that the attribute descriptions simply need to be the descriptive attribute names themselves which are present in all databases.
>
> We provide some examples of the type of descriptions expected: prescriptions-dose_unit_rx, noteevents-text, procedureevents_mv-location. We can see these are not incredibly descriptive (i.e. they are simply meaningful names) but an LLM can contextualize and reason about these simple descriptions. Of course, if the attribute names are completely uninformative (e.g. X1, X2, X3) then this would be suboptimal for Matchmaker.
>
> We also highlight that this links to our rationale behind the multiple approaches to candidate generation (semantic and reasoning). As shown in Section 5.3, our reasoning-based candidates often outperform pure semantic matching, highlighting resilience to description quality. We hope this clarifies.
>
> - ``Q4: Strategies to Reduce LLM Calls``
>
> We have implemented several strategies to optimize the number of LLM calls, especially in contrast to the related work which often does pairwise assessment (Cartesian product) of source-target attributes. We can see the reduced calls of Matchmaker vs baselines as shown in Table 6 (Appendix D.1)
>
> Specifically, Matchmaker achieves this efficiency through several key design choices:
> (1)   the retrieval-based candidate generation significantly reduces the search space compared to exhaustive comparison approaches,
> (2)   focused reasoning on refined candidate sets makes optimal use of more expensive LLM operations.
> (3)   The MCQ format enables efficient single-pass confidence scoring, further reducing computational overhead.
>
> These advantages directly translate to practical benefits to reduce the number of LLM calls.
>
> - ``Q5: Handling Schemas in Non-English Languages``
>
> Regarding multi-lingual capabilities, while our evaluation uses English schemas as they represent available benchmarks with expert validation, Matchmaker's approach is inherently language-agnostic by design.
>
> The high-level architectural approach would hence remain unchanged for different languages, however, two changes might be needed- either translation of schemas in other languages to English (pre-processing) or using LLMs which are inherently multi-lingual. We thank the reviewer for raising this and is a promising future investigation — especially if multi-lingual datasets become available.
>
> **UPDATE:** We have updated the discussion to include this.
>
> ----
> _Thank you for your help in improving our work. We hope these answer your points, please let us know if there are any remaining concerns!_

---

> > ### Author Response · Authors · 2024-11-22
> >
> > Dear ``Reviewer JXrD``
> >
> > We are sincerely grateful for your time and efforts in the review process.
> >
> > We hope that our responses have been helpful. Given the limited time left in the discussion period, please let us know if there are any leftover concerns and if there is anything else we could do to address any further questions or comments. We are looking forward to your further feedback!
> >
> > Thank you!
> >
> > Paper 7911 Authors

---

> > > ### Author Response · Authors · 2024-11-25
> > >
> > > Dear Reviewer ``JXrD``
> > >
> > > We would like to express our sincere gratitude for the time and effort you have dedicated to providing thoughtful feedback to improve our paper.
> > >
> > > The process of addressing your comments has led to significant updates to the paper, and we are grateful for the opportunity to improve it based on your guidance.
> > >
> > > We hope the reviewer agrees that the improvements based on your suggestions have improved the paper, compared to the earlier draft. With this in mind, we kindly ask you to please consider whether the score assigned to the earlier version reflects the current version paper. _We would greatly appreciate it if you might reconsider raising your score based on the improved paper._
> > >
> > > Thank you once again for your constructive suggestions, which have made the paper all the better (thank you!)
> > >
> > > Paper 7911 Authors

---

> > > > ### Comment · Reviewer_JXrD · 2024-11-26
> > > >
> > > > Thank you for the comprehensive responses to my review. The authors have thoroughly addressed all concerns raised in the initial review through detailed explanations and additional experimental work. Specifically:
> > > >
> > > >   1.  The justification for focusing on healthcare datasets (MIMIC and Synthea) is well-reasoned, Highlighting both their role as standard benchmarks and their representation of real-world schema matching challenges.
> > > >   2.  The synthetic in-context example generation process has been clarified substantially, with a new detailed explanation in Appendix A.6 that effectively addresses the previous ambiguity around the self-improvement mechanism.
> > > >   3.  The authors have conducted new error analysis experiments, documenting that 83% of errors involve semantically related attributes (mean similarity of 0.862), with results now detailed in Appendix D.6.
> > > >   4.  All technical questions regarding scalability, many-to-many mappings, sensitivity to descriptions, LLM call optimization, and multi-lingual capabilities have been addressed comprehensively, with several improvements incorporated into the discussion section.
> > > > The additional experimental results and manuscript updates have further strengthened what was already a good paper. The original rating of 8 (accept) is firmly supported by these improvements.

---

> > > > > ### Author Response · Authors · 2024-11-26
> > > > >
> > > > > Dear Reviewer JXrD
> > > > >
> > > > > Thank you for time and your positive assessment of our paper! We greatly appreciate your suggestions which have helped us to strengthen it.
> > > > >
> > > > > Paper Authors

---

### Official Review · Reviewer_N216 · 2024-11-03

**Soundness:** 2
**Presentation:** 1
**Contribution:** 2
**Rating:** 3
**Confidence:** 4

**Summary:**

In this paper, the authors considered the schema matching problem for
tabular structured data. In particular, a schema is defined as a set
of tables {T_1, ..., T_m}, where each table T_i has a set of
attributes {A_{i,1}, ..., A_{i,n_i}}. Additionally, it is assumed that
each table T_i is associated with some metadata describing its purpose
and content, and each attribute A_{i,j} is associated with some
metadata describing its type and relational context. Then, given a
source schema S including a set of attributes A_S and a target schema
T including a set of attributes A_T, the goal of schema matching is to
find a partial function f : A_S -> A_T such that if f(A) = B for
attributes A in A_S and B in A_T, then B is the corresponding target
attribute to the source attribute A.

The algorithm proposed in the paper for schema matching works as
follows. Given a source attribute A, the algorithm uses embeddings of
A and the target attributes to retrieve the top-k matching target
attributes. The generated set of target attributes is called the
semantic retrieval candidates for A. Then the algorithm generates a
set of reasoning-based candidates using a reasoning LLM. The union of
semantic retrieval candidates with reasoning-based candidates
constitutes the set of candidates for A. Then the algorithm uses a
refiner LLM to reduce the number of candidates, and it finally ranks
the resulting candidates and filters out the non-suitable ones. If the
resulting set of target attributes is not empty, then the top-scored
attribute can be considered as the corresponding target attribute for
A.

**Strengths:**

S1) The schema matching problem is fundamental to generating large,
    integrated, and interoperable datasets. In this sense, this paper
    addresses a relevant and interesting problem.

S2) The experimental evaluation shows that the proposed approach
    outperforms other schema matching approaches in terms of accuracy.

**Weaknesses:**

W1) The approach proposed in the paper essentially disregards the
    large body of work on schema matching that has been developed for
    decades in the database field.

W2) The schema matching problem is not properly formalized.

**Questions:**

General comments

The schema matching problem is fundamental to generating large,
integrated, and interoperable datasets. In this sense, this paper
makes a contribution by proposing a new method for this problem that
takes advantage of certain capabilities of LLMs. Besides, the
experimental evaluation provides evidence that the proposed method
outperforms other schema matching approaches in terms of
accuracy. However, I have two serious concerns about the contribution
of this paper:

- The schema matching problem is not properly formalized. The authors
  provide a formal definition of this problem where they indicate that
  mapping function f "correctly" assigns each attribute of the source
  schema to an attribute of the target schema. How is the
  "correctness" of f defined? What are the properties that a correct
  mapping function f should satisfy? Does the algorithm proposed in
  this paper compute a function f that satisfies such properties? None
  of these questions is answer in the paper.

- The authors disregard the large body of work on schema matching that
  has been developed in the database area. I am not going to mention
  here the relevant literature on schema matching, which is an
  established area within databases, but I would like to note that one
  can already find surveys on this topic from more than 20 years ago:

  Erhard Rahm, Philip A. Bernstein: A survey of approaches to
  automatic schema matching. VLDB J. 10(4): 334-350 (2001)

  Notice that the problem considered in this paper is schema matching
  for relational databases, which is exactly the scenario discussed in
  this survey.

  A first obvious step in considering the work on schema matching done
  in databases is to compare the method proposed in this paper with
  methods from the database field. But this is just the tip of the
  iceberg and probably not the most fruitful way. The method proposed
  in this paper can be improved by considering the more classical work
  in schema matching. For example, the authors could leverage this
  work in the generation of candidates, and they can address issues
  with the formalization of the schema matching problem by reusing its
  formalization from the database field.


Specific questions

All these questions refer to the definition of the schema matching
problem:

- How is the notion of correctness of a mapping function f defined?

- What are the properties that a correct mapping function f should
  satisfy?

- Does the algorithm proposed in this paper compute a function f that
  satisfies such properties?

---

> ### Author Response · Authors · 2024-11-18
> **Response to Reviewer N216 [Part 1/3]**
>
> Dear ``Reviewer N216``
>
> Thank you for your thoughtful comments and suggestions! We give answers to each of the following in turn, along with corresponding updates to the revised manuscript.
>
> (A) Related work: Classical schema matching literature. [**PART 2/3**]
>
> (B) Formalism clarifications: defining correctness & properties $f$ should satisfy. [**PART 3/3**]
>
> (C) Formalism clarifications: how does Matchmaker satisfy the properties. [**PART 3/3**]

---

> ### Author Response · Authors · 2024-11-18
> **Response to Reviewer N216 [Part 2/3]**
>
> # **(A) Related work: Classical schema matching literature**
>
> Thank you for your thoughtful feedback and for highlighting the survey by Rahm and Bernstein (2001). We acknowledge the rich history of schema matching within the database community as we highlighted in Section 2, L131-138, where we discussed earlier schema matching works and noted their focus on simpler schemas and entity matching rather than complex multiple table schema matching. In addition to LLM baselines, we also selected SMAT as a baseline because it is the SOTA method (pre-LLM) and has been shown on the benchmark datasets to outperform the classical approaches, establishing itself as a robust baseline.
>
> We appreciate the opportunity to expand upon Matchmaker's place within the broader literature and to contextualize our contributions more clearly. Based on your feedback we have made the following revisions to our related work to better contextualize & contrast Matchmaker with earlier schema matching approaches.
>
> **Classical Schema Matching approaches**
>
> Classical schema matching uses a range of strategies, including heuristic-driven linguistic matching, constraint-based methods, and structural analysis. These methods have focused on simple schemas (individual tables), performing entity matching (see **L131-138**).
>
> ---
>
> ### **Weaknesses of Classical Approaches vs Matchmaker:**
>
> - **Single-Table/Simple schema:** Classical methods typically perform schema matching at the element level, treating tables as isolated entities and matching attributes based on direct comparisons of names, data types, or simple structural cues. In particular, often a focus was simple schemas, where the goal was to map elements between single tables. However, this approach fails to handle the complexity of modern data systems, where schemas are often multi-table, hierarchical, or require cross-table reasoning.
>
> **Contrast:** Matchmaker uses LLM-based reasoning to connect attributes across multi-table and hierarchical schemas, understanding how data relationships span multiple tables. This permits handling complex and interrelated schema structures.
>
> - **Dependency on Heuristics and Limited Semantic Understanding:** Classical methods rely on heuristic-driven matching based on linguistic similarities (e.g., name matching using synonyms, hypernyms, or edit distance) and structural constraints like key relationships​. While these heuristics work in well-defined contexts, they are insufficient for domains where semantic meaning is implicit, such as in healthcare and as per Fig 1 --- only semantic matching is insufficient.
>
> **Contrast:** Matchmaker employs chain-of-thought with LLMs to perform reasoning, allowing it to capture relationships that are not explicitly defined in the schema structure or names. This enables Matchmaker to handle complex mappings that classical methods cannot infer.
>
> - **Manual Effort and Lack of Adaptability:** Classical techniques require significant manual effort for tuning and adaptation, which means they aren’t zero-shot. Constraint-based approaches, in particular, need manual intervention​.  Alternatively, they might also rely on labeled data for effective matching. This makes these classical approaches impractical in the real-world.
>
> **Contrast:** Matchmaker’s zero-shot and self-optimization capabilities mean it can adapt autonomously to new schemas using synthetic in-context examples, significantly reducing the need for manual tuning and making it more practical for dynamic, real-world data integration tasks.
>
> ---
>
> ### **Key Weaknesses of SMAT and How Matchmaker Addresses Them:**
>
> We also compared Matchmaker to state-of-the-art (SOTA) methods like SMAT (Zhang et al., 2021), which applies attention mechanisms for schema matching​. While SMAT represents an important advancement over classical methods, it has several limitations that Matchmaker overcomes:
>
> - **Dependency on Labeled Data:** SMAT requires extensive labeled data (over 50% labeled matches) for training, which is often impractical in real-world schema matching.
>
> **Contrast:** Matchmaker’s zero-shot matching capability allows it to perform well without any labeled training data, using LLMs to generate and refine matches autonomously.
>
> - **Binary formulation:** SMAT formulates the problem as a binary classification task over the full Cartesian product of source and target schema attributes. e.g. for each pair of source-target attributes. This leads to a large amount of comparisons.
>
> **Contrast:** Matchmaker’s formulation as information retrieval reduces the number of comparisons and leads to greater efficiency --- in addition to better performance.
>
> ---
>
> **UPDATE:** Expanded the Related Work section to discuss the above works based on the Rahm & Bernstein paper (which is now cited to provide readers with the context). See Appendix A.7.
>
> Thank you for your valuable suggestions, which have significantly strengthened our manuscript.

---

> ### Author Response · Authors · 2024-11-18
> **Response to Reviewer N216 [Part 3/3]**
>
> ## **(B) Formalism clarifications: defining correctness & properties $f$ should satisfy**
>
> In practice, correctness in schema matching is evaluated against expert-validated ground truth mappings between source and target schemas (e.g. MIMIC to OMOP and Synthea to OMOP).
>
> Based on your question, we have added dimensions that represent what goes into defining this human-annotated notion of matching correctness and hence what properties $f$ should possess. These lie along the following dimensions
>
> - Semantic Equivalence/Consistency: $f(A_ₛ) = A_ₜ$ implies $A_ₛ$ and $A_ₜ$ represent the same real-world concept (i.e. the mapped attributes serve equivalent purposes)
> - Type Compatibility: Mapped attributes must have compatible data types.
> - Structural Consistency: Mappings must respect schema hierarchies.
> - Coverage: $f$ should identify valid matches while avoiding incorrect mappings through abstention. i.e. coverage is maximized by improved accuracy@k
>
> We can then practically assess if a function $f$ (such as Matchmaker) satisfies these criteria based on its performance against expert-validated ground truth mappings in real-world benchmark datasets as has been done in the paper.
>
> Thank you for the suggestion which has improved the paper.
>
> **UPDATE:** We have included a new Appendix A.5 to include this discussion around the properties of schema matching and flag this in the main paper.
>
>
> ----
>
> ## **(C) Formalism clarifications: how does Matchmaker satisfy the properties?**
>
> Yes, as empirically shown Matchmaker best satisfies the properties needed of a schema matching function $f$, based on its strong performance on real-world schema matching tasks where it significantly outperforms
> existing approaches on standard benchmarks (Table 2). In particular, the strong empirical performance outperforming baselines implies that Matchmaker better satisfies the properties as compared to baseline schema matching algorithms
>
> Matchmaker also has specific design aspects within its compositional LLM structure that promote addressing the properties.
>
> - (1)   Semantic equivalence/consistency: Matchmaker employs multiple mechanisms: multi-vector document representation captures semantic nuances beyond simple name matching, while dual candidate generation combines both semantic retrieval and LLM reasoning to identify conceptually equivalent attributes.
> - (2)   Type compatibility: enforced through the inclusion of data type information in our multi-vector documents (Section 4.1) and LLM reasoning during candidate generation and refinement (Section 4.2), with examples in Appendix C showing explicit consideration of type compatibility (e.g., string->varchar, integer->bigint).
> - (3)   Structural consistency is maintained by incorporating table metadata and hierarchical information in document creation (Section 4.1), using reasoning-based candidate generation that considers schema structure (Section 4.2), and including table context in confidence scoring.
> - (4)   Coverage is optimized through our MCQ format with a "None of the above" option enabling abstention when no good match exists, while confidence scoring helps identify and rank high-quality matches. This property does not exist in any of the baselines.
> Consequently, our empirical results validate that these properties then translate to superior performance in practice.
>
> **UPDATE:** We have included a new Appendix A.5 to include this discussion of how Matchmaker satisfies the properties.
>
>
>
> ----
> _Thank you for your help in improving our work. We hope these answer your points, please let us know if there are any remaining concerns!_

---

> > ### Author Response · Authors · 2024-11-22
> >
> > Dear ``Reviewer N216``
> >
> > We are sincerely grateful for your time and efforts in the review process.
> >
> > We hope that our responses have been helpful. Given the limited time left in the discussion period, please let us know if there are any leftover concerns and if there is anything else we could do to address any further questions or comments. We are looking forward to your further feedback!
> >
> > Thank you!
> >
> > Paper 7911 Authors

---

> > > ### Author Response · Authors · 2024-11-27
> > >
> > > Dear ``Reviewer N216``
> > >
> > > We would like to express our sincere gratitude for the time and effort you have dedicated to providing thoughtful feedback which has helped us to update and improve the paper. We believe in addressing your concerns that the paper is all the better for it (thank you!)
> > >
> > > We hope the reviewer agrees that the improvements based on your suggestions have improved the paper, compared to the earlier draft. With this in mind, we kindly ask you to please consider whether the score of 3 (reject) assigned to the earlier version reflects the current version of the paper. We would greatly appreciate it if you might reconsider raising your score based on the improved paper.
> > >
> > > Thank you once again for your constructive suggestions, which have improved the paper.
> > >
> > > Paper 7911 Authors

---

> > > > ### Author Response · Authors · 2024-12-02
> > > >
> > > > Dear ``Reviewer N216``
> > > >
> > > > Regarding the original review, we hope the responses and paper updates based on your suggestions have addressed the your concerns.
> > > >
> > > > We hope the reviewer agrees that the improvements based on your suggestions have improved the paper, compared to the earlier draft. With this in mind, we kindly ask you to please consider whether the score of 3 (reject) assigned to the earlier version reflects the current version of the paper. We would greatly appreciate it if you might reconsider raising your score based on the improved paper.
> > > >
> > > > Given the limited time left in the discussion period, please let us know if there are any leftover concerns and if there is anything else we could do.
> > > >
> > > > Paper 7911 Authors

---

### Official Review · Reviewer_TBJ4 · 2024-11-03

**Soundness:** 3
**Presentation:** 4
**Contribution:** 2
**Rating:** 5
**Confidence:** 4

**Summary:**

This paper presents a zero-shot schema matching approach by leveraging the multi-stage call of LLMs to generate, refine, and score the matchings.  Specifically, it introduces synthetic examples to guide the reasoning of LLMs for improving and optimizing the results of schema matching. The experiments on medical schema matching benchmarks demonstrate that the proposed approach outperforms the selected baseline methods on accuracy.

The paper does a great job of demonstrating the problem that they are solving and the methodology they presented.
The main contribution is decomposing the schema matching task into multi-stage sub-tasks that are completed by multiple calling of LLMs, with retrieval from contextual reasoning and prompt optimization based on in-context examples. However, the contribution of this work is limited, as they only introduce the muti-stage schema matching by extending the calling of LLMs from single to multiple.

**Strengths:**

S1: The idea of leveraging the multi-stage LLMs for schema matching is novel.

S2: The authors do a great job of demonstrating the challenges of schema matching in real-world scenarios that they are trying to address and the methodology they presented.

S3: Several experiments on MIMIC-OMOP and Synthea-OMOP datasets are conducted to empirically investigate and demonstrate the performance of the presented method.

**Weaknesses:**

W1: The contribution of this work is limited, as they only introduce the muti-stage schema matching by extending the calling of LLMs from single to multiple.

W2: The accuracy@k is the only metrics that is reported in experimental results, the results on precision are missing.

W3: The prompts are provided in the appendix, but the source code is not provided for reproducibility.

W4: The GPT-4 (0613) is the only backbone model, the results of using llama as backbone are not reported.

**Questions:**

Q1: How does your approach work in terms of precision when compared with the baseline method?

Q2: How confident when ranking the LLM-generated candidates with LLM-based scores? How much does this ranking contribute to the results?

Q3: Could you provide the details of how the vector retrieval works in Sec 4.1?
If I understand well, you retrieve the top-k matching from the target schema attribute based on MaxSim between query embeddings and target schema embeddings. However, the granularity of the query embeddings and target schema embedding is different, query embedding is an attribute-level embedding while target schema embedding is a table-level embedding.

---

> ### Author Response · Authors · 2024-11-18
> **Response to Reviewer TBJ4 [Part 1/2]**
>
> Dear ``Reviewer TBJ4``
>
> Thank you for your thoughtful comments and suggestions! We give answers to each of the following in turn, along with corresponding updates to the revised manuscript.
>
>
> (A) Clarifying novelties/contributions of Matchmaker [**PART 1/2**]
>
> (B) Performance metrics - precision [**PART 1/2**]
>
> (C) Clarifying backbone LLM [**PART 2/2**]
>
> (D) Questions [**PART 2/2**]
>
>
>
> ----
>
> ## **(A) Clarifying novelties/contributions of Matchmaker**
>
> We respectfully disagree that Matchmaker’s contribution is only  ‘’extending the calling of LLMs from single to multiple’’. We would like to clarify the different technical novelties, as well as, the formulation contribution which we believe has been overlooked.
>
> **(1) Self-Improvement via Synthetic In-Context Examples:** Our unique contribution is our zero-shot self-improvement mechanism for compositional LLM systems. Specifically, Matchmaker optimizes in a zero-shot manner via synthetic in-context examples that lead to performance improvements without labeled data. This mechanism systematically selects in-context examples for different stages, allowing for the enhancement of compositional LLM programs without labeled data. We note this contribution applies beyond schema matching to any compositional LLM program. This specific novelty sets Matchmaker apart from existing methods that either require extensive labeled data or lack a mechanism for self-improvement.
>
> **(2) Confidence Scoring for Human-in-the-Loop Deferral:** Matchmaker introduces principled uncertainty quantification through MCQ-based confidence scoring. This contribution is vital for practical deployment as it enables Matchmaker to identify high-uncertainty cases and defer them to human experts when uncertain (i.e. human-in-the-loop deferral). This capability is absent from existing schema matching approaches.
>
> **(3) Multi-Stage Compositional LLM Program**: Unlike traditional methods that perform schema matching in a single step, Matchmaker defines a novel compositional LLM program comprising candidate generation, refinement, and confidence scoring. This novel compositional LLM program enables iterative reasoning and improves performance in complex schema matching scenarios.
>
> **Novel formulation for schema matching:** As discussed in Sec 3.2, Matchmaker also proposes a novel formulation of schema matching as an information retrieval problem rather than binary classification as is done in prior schema matching works, significantly improving scalability.
>
> **UPDATE:** We have updated our contributions section of the paper to better clarify.
>
>
>
> ----
>
> ## **(B) Performance metrics - precision**
>
> We fully agree with the reviewer that precision is an important metric to consider. We clarify that in our m:1 schema matching setting, **accuracy@1 and precision are mathematically equivalent**.
>
> This equivalence holds because our model assigns matches using argmax - that is, for each source attribute, we select the target attribute with the highest confidence score as the predicted match. Since each source attribute receives exactly one prediction (due to argmax) and can only match to one target attribute (the m:1 constraint), precision (TP/(TP+FP)) becomes identical to accuracy@1 (number of correct matches / total number of predictions).
>
> Therefore, our strong accuracy@1 results directly demonstrate Matchmaker's precision advantages over the baselines.
>
> **UPDATE**: Clarified this in the updated Sec 5 of the paper.

---

> ### Author Response · Authors · 2024-11-18
> **Response to Reviewer TBJ4 [Part 2/2]**
>
> ## **(C) Clarifying backbone LLM**
>
> To clarify our choice of GPT-4 as the LLM backbone is to ensure consistency and fair comparison with the LLM baseline methods which use GPT-4. This ensures that our performance improvements are attributable to the methodological and architectural contributions of Matchmaker, rather than the choice of LLM itself.
>
> That said, we highlight in Appendix D.2, we conduct experiments with other LLMs. Naturally, the capability of the LLM influences Matchmaker’s results (i.e. the better the LLM, the better Matchmaker performs).
> However, to give an example when we use GPT-3.5 as the Matchmaker backbone we still find it outperforms our closest baseline ReMatch (when ReMatch uses GPT-4).
>
> This highlights that Matchmaker’s technical contributions are a key source of performance gain, rather than just the choice of LLM itself.
>
> **UPDATE:**  We have updated Sec 5.1  to better clarify our use of GPT-4 ensures fair comparison with the baselines and isolate the system gains not tied to the LLM itself.
>
>
> ----
>
> ## **(D) Questions**
>
> - ``Q1: How Matchmaker performs in terms of precision``
>
> Please see the answer in (B)
>
> - ``Q2: Contribution of ranking to Matchmaker’s results.``
>
> Our confidence scoring and ranking are important aspects of Matchmaker’s performance. As shown in Fig 4 (a) deferral based on the entropy of the ranked prediction set can lead to significant performance gains in accuracy@1. This demonstrates the impact of confidence scoring on performance, particularly in human-in-the-loop scenarios.
>
> **New experiment:** Additionally, we have conducted a new experiment, where we ablate the ranking step.
>
> The results shown below highlight the importance of the re-ranking step towards achieving better accuracy@1. Thank you for the suggestion!
>
> |         |       | Matchmaker (with ranking) | Matchmaker (No ranking) |
> |---------|-------|---------------------------|-------------------------|
> | MIMIC   | Acc@1 | 62.20                     | 57.00                   |
> |         | Acc@3 | 68.80                     | 66.90                   |
> |         | Acc@5 | 71.10                     | 71.10                   |
> |---------|-------|---------------------------|-------------------------|
> | Synthea | Acc@1 | 70.20                     | 62.40                   |
> |         | Acc@3 | 78.60                     | 77.20                   |
> |         | Acc@5 | 80.90                     | 80.90                   |
>
> **UPDATE:** Add the new experiment (ablation) to Appendix D.7 of the updated paper.
>
>
> - ``Q3: Clarifying details on Vector Retrieval``
>
> To clarify the process of vector retrieval. We treat each target schema table as a "document," and the attributes within the table serve as the document's contents. Using ColBERT, we generate token-level embeddings for these documents, which are indexed for efficient retrieval. The query (an individual attribute from the source schema), is then represented using token-level embeddings via ColBERT. This ensures a consistent token-level embedding structure across both query attributes and target schema documents.
> Thereafter, when matching the query embeddings are compared against the token-level embeddings of each document in the index. The late interaction mechanism computes the MaxSim scores for each document, identifying the most relevant token matches for the query. Finally, for each document, MaxSim identifies the highest similarity scores for the query tokens, and these scores are aggregated to generate a relevance score for that document. All documents are then ranked based on their overall relevance scores. The top-k documents, which contain the most semantically similar attributes to the query, are retrieved as matches.
>
> **UPDATE:** We have updated Sec. 4.1 and 4.2 to better clarify the vector retrieval
>
> - Clarifying code release:
> We agree with the reviewer and note that the code will be released upon acceptance, along with detailed documentation.
>
> ----
>
> _Thank you for your help in improving our work. We hope these answer your points, please let us know if there are any remaining concerns!_

---

> > ### Author Response · Authors · 2024-11-22
> >
> > Dear Reviewer TBJ4
> >
> > We are sincerely grateful for your time and efforts in the review process.
> >
> > We hope that our responses have been helpful. Given the limited time left in the discussion period, please let us know if there are any leftover concerns and if there is anything else we could do to address any further questions or comments. We are looking forward to your further feedback!
> >
> > Thank you!
> >
> > Paper 7911 Authors

---

> ### Author Response · Authors · 2024-11-25
> **Response: Reviewer TBJ4**
>
> Dear Reviewer TBJ4
>
> Thank you for your response. We are glad we could address your concerns with our improvements and clarifications on novelty.
>
> We wish to clarify the remaining two concerns & have uploaded a revised manuscript to further ensure clarity:
> (i) Clarifying Metrics and (ii) LLM ablation
>
>
> **(i) Clarifying Metrics:**
>
> Thank you for your follow-up comment regarding acc@1, precision and F1-score. We appreciate the opportunity to clarify and address this point in more detail.
>
> In our _specific m:1_ schema matching setting, where each source attribute receives exactly one prediction (due to the argmax mechanism) and can only match to one target attribute, acc@1 is indeed equivalent to both precision and recall. This equivalence arises because:
>
> - Precision (TP / (TP + FP)): In our setup, every prediction is evaluated, and there are no unassigned predictions. Thus, the number of true positives (TP) is the same as the number of correct matches, and since there are no unassigned or extraneous predictions, precision equals the fraction of correct matches, which is captured by accuracy@1.
>
> - Recall (TP / (TP + FN)): In our setting, every source attribute must be matched, meaning there are no false negatives. Consequently, recall is also identical to accuracy@1.
>
> - F1-Score: As the harmonic mean of precision and recall, F1-score becomes equivalent to accuracy@1 in this setup, since precision and recall are the same.
>
> We realize that our earlier explanation could have caused some confusion, and we thank the reviewer for pointing this out. To explicitly clarify:
>
> - Accuracy@1 is equivalent to both precision and recall in our setting.
> - F1-score is also equivalent to accuracy@1, as precision and recall are identical in this setup.
>
> **UPDATE:** We have revised the manuscript to include a **new Appendix on Metrics (Appendix A.8)** highlighting this and have flagged this in the main paper (**L404**). We hope this update and explanation resolves any ambiguity.
>
> ---
>
> **(ii) LLM Ablation:**
>
> We wish to clarify as noted on **L417** that we ablate other LLMs in Appendix D.2.
>
> As expected, Matchmaker's results are influenced by using a weaker LLM (i.e. the better the LLM, the better Matchmaker performs). However, what this ablation highlights is that even with a weaker backbone LLM we still outperform our closest baseline ReMatch (when ReMatch uses GPT-4 - stronger LLM). This highlights that Matchmaker’s technical contributions are a key source of performance gain, rather than just the choice of LLM itself.
>
> **UPDATE:** We have updated the main paper **L416-417** to more clearly flag the ablation.
>
> ----
> We appreciate the reviewer’s thoughtful feedback on metrics and the LLM ablation study, which allowed us to clarify and further strengthen the manuscript. We are especially grateful for the reviewer’s valuable suggestions, which directly contributed to key improvements in our paper, including earlier enhancements to better convey our novelty and motivate new experiments.
>
> _We understand that the current score of 5 was based on the earlier version of the paper. Based on your feedback, we have substantially updated the manuscript with additional clarifications and improvements. We hope the reviewer will agree that these updates, guided by your suggestions, have resulted in a much stronger paper. We would greatly appreciate it if you might reconsider your score based on the improved paper._
>
> Thank you for your suggestions which have helped us improve the paper!
>
> Paper 7911 Authors

---

> ### Author Response · Authors · 2024-11-26
>
> Dear ``Reviewer TBJ4``,
>
> Thank you for your detailed and thoughtful feedback. We wish to clarify your remaining concerns.
>
> ---
>
> ## **Scope of Mappings**
> As noted in L402, our work focuses on m:1 mappings (which encompass many-to-one and one-to-one). This is consistent with established benchmarks in the literature, such as ReMatch and SMAT, which also target m:1 mappings as their focus. Like these previous works, we limit our analysis to evaluating 1:1 or m:1 relations, reflecting the dominant scenarios in the datasets we study. Our goal in aligning our scope with these state-of-the-art methods was to ensure comparability to these baselines.
>
> We thank the reviewer for clarifying the importance of one-to-many (1:many) and many-to-many (m:n) mappings. As noted in our discussion, future work should extend the approach to handle these cases.
>
> _That said, we wish to highlight that m:1 mappings are vital and have multiple use cases in four different industry schema-matching tasks._
>
> **Real-World relevance of Matchmaker targeting m:1 mappings (with examples from different industries)**:
>
> Below, we provide concrete examples of the critical importance of m:1 mappings studied in our paper across four different real-world domains/industries:
>
> (1) Healthcare Applications:
>
> - Clinical Events:  In electronic health records (EHRs), attributes like “starttime” and “endtime” of a medical procedure often map to a single target attribute like “procedure_duration” in standardized schemas such as OMOP.
> - Medication Data Integration: Attributes like "medication_name," "dosage," and "frequency" can map to a single "medication_record" target attribute.
> - Patient Demographic measures: Separate attributes like "city" and "state" might map to a unified "location" field.
>
> (2) Financial Services:
>
> - Transaction Reconciliation: Attributes like "credit_amount" and "debit_amount" often map to a single "transaction_balance" attribute in banking systems.
> - Risk Assessment: In insurance, data fields like "accident_severity" and "number_of_claims" might map to a single "risk_score" attribute.
>
> (3) E-Commerce and Logistics:
>
> - Order Consolidation: Fields such as "item_cost," "shipping_fee," and "tax" might map to a single "total_price" attribute.
> - Delivery Information: Attributes like "pickup_date" and "delivery_date" often map to a "shipment_duration" field.
>
> (4) Energy Sector:
>
> - Sensor Data Aggregation: In smart grids, attributes such as "voltage_reading" and "current_reading" can map to a single "power_consumption" attribute.
>
> These examples demonstrate that m:1 mappings are not only frequent but also essential to enable interoperability in diverse domains. This highlights the relevance of schema-matching methods like Matchmaker addressing these challenges.
>
> **Alignment with Figure 1**:
> Finally, to clarify Figure 1 in our manuscript explicitly illustrates mappings to a single target element, aligning with our formalism of m:1 mappings.
>
> ---
>
> ## **Backbone LLM**
>
> We appreciate the reviewer’s interest in the results of different backbone LLMs. As detailed in Appendix D.2, we conducted ablations with GPT-3.5 & GPT-4. This reflects LLMs with different levels of capabilities (and parameter counts). Our findings demonstrate:
>
> - Stronger LLMs improve performance, but the gains are primarily driven by Matchmaker’s compositional reasoning and optimization strategies rather than the choice of LLM alone.
> - Even with a weaker backbone LLM, Matchmaker consistently outperforms the closest baseline, ReMatch, highlighting the robustness and adaptability of our method.
>
> To emphasize this analysis as suggested by the reviewer, we have added further details from the ablation study to the main text (L417) and clarified the results' implications for broader applicability.
>
> ---
>
> We sincerely thank the reviewer for raising these points, which allowed us to further clarify our work.
> We hope these updates address your concerns and further highlight the contributions of our paper. Please let us know if there are any leftover concerns and if there is anything else we can do. We are looking forward to your further feedback!
>
> Thank you
>
> Paper 7911 Authors

---

### Official Review · Reviewer_fXFk · 2024-11-04

**Soundness:** 3
**Presentation:** 4
**Contribution:** 2
**Rating:** 5
**Confidence:** 4

**Summary:**

The paper deals with schema matching, and old but very important problem in databases. The idea is that, given one starting (relational) schema and one target schema, to be able to match which attributes in the  starting schema correspond to attributes in target schema. The proposal in this paper is Matchmaker. This system uses a mix of retrieval using multi-vector representation and LLM-based reasoning to produce candidates for the matching, and then applies a final LLM-driven step to refine these candidates. Notably, the program also is built so that it can optimize the last step by providing examples from the databases. All together, the system shows quite an advantage over previous proposals.

**Strengths:**

- Very well written
- Addresses an importan tproblem that has ben recently identified as a target for the ML community
- Provides a thorough experimental section, including a (very nice) ablation study to understand the impact of different strategies for candidate generation.

**Weaknesses:**

- While the paper is well writtend and scientifically sound, the algorithm itself (matchmaker) is not groundbreaking. Matchmaker resolves basically on building appropriate chain of thought prompts, as well as applying semantic similarity techniques. As such, I see this mostly as a paper describing a particular, LLM-based proposal, to address this problem.
- There seem to be a lack of LLM-dirven alternatives to compare with, which is both good for the paper (because authors are the first to apply them in this context), but it also raises the question on whether any other similar approach would produce similar results.
- The problem itself ( schema matching, or more generally data harmonization/interoperability) is not a core problematic of ICLR. I would imagine this paper would be more suitted to a database conference.

**Questions:**

Why haven't you submitted this to a database conference? It seems to me that the reception and impact there would be much higher.

---

> ### Author Response · Authors · 2024-11-18
> **Response to Reviewer fXFk [Part 1/2]**
>
> Dear ``Reviewer fXFk``
>
> Thank you for your thoughtful comments and suggestions! We give answers to each of the following in turn, along with corresponding updates to the revised manuscript.
>
> (A) Clarifying novelties/contributions of Matchmaker [**Part 2/2**]
>
> (B) Clarifying our existing LLM baseline comparisons [**Part 2/2**]
>
> (C) Clarifying why Matchmaker fits ICLR [**Part 2/2**]
>
> ----

---

> ### Author Response · Authors · 2024-11-18
> **Response to Reviewer fXFk [Part 2/2]**
>
> ## **(A) Clarifying novelties/contributions of Matchmaker**
>
> We respectfully disagree that Matchmaker’s contribution and novelty is just ‘’chain of thought prompts and applying semantic similarity techniques’’. We would like to clarify the different technical novelties, as well as, the formulation contribution which we believe has been overlooked.
>
> **(1) Multi-Stage Compositional LLM Program**: Unlike traditional methods that perform schema matching in a single step, Matchmaker defines a novel compositional LLM program comprising candidate generation, refinement, and confidence scoring. This novel compositional LLM program enables iterative reasoning and improves performance in complex schema matching scenarios.
>
> **(2) Confidence Scoring for Human-in-the-Loop Deferral:** Matchmaker introduces principled uncertainty quantification through MCQ-based confidence scoring. This contribution is vital for practical deployment as it enables Matchmaker to identify high-uncertainty cases and defer them to human experts when uncertain (i.e. human-in-the-loop deferral). This capability is absent from existing approaches.
>
> **(3) Self-Improvement via Synthetic In-Context Examples:** Our unique contribution is our zero-shot self-improvement mechanism for compositional LLM systems. Specifically, Matchmaker optimizes in a zero-shot manner via synthetic in-context examples, leading to performance improvements without labeled data. This mechanism systematically selects in-context examples for different stages, allowing for the enhancement of compositional LLM programs without labeled data. We note this contribution applies beyond schema matching to any compositional LLM program. This specific novelty sets Matchmaker apart from existing methods that either require extensive labeled data or lack a mechanism for self-improvement.
>
> **Novel formulation for schema matching:** As discussed in Sec 3.2, Matchmaker also proposes a novel formulation of schema matching as an information retrieval problem rather than binary classification as is done in prior schema matching works, significantly improving scalability
>
> **UPDATE:** We have updated our contributions section of the paper to better clarify.
>
> ----
>
> ## **(B) Clarifying our existing LLM baseline comparisons**
>
> We clarify that our evaluation **does already** include comparisons against multiple LLM-driven approaches spanning different paradigms (see **L414-420**).
>
> Specifically, we evaluate Matchmaker vs the following LLM-driven baselines:
> - ReMatch (retrieval-augmented LLM approach),
> - LLM-DP (pure prompting-based approach), and
> - Jellyfish (both 7b and 13b LLM variants which are specifically fine-tuned for data preprocessing tasks).
>
> As shown in **Table 2**, Matchmaker consistently outperforms these LLM-based methods, as demonstrated by our experiments on real-world benchmarks. These results validate that our technical innovations provide meaningful benefits beyond existing LLM-driven methods.
>
> ----
>
> ## **(C) Clarifying why Matchmaker fits ICLR**
>
> While schema matching originated in the database community, we believe submission to an ML venue such as ICLR is important for the following four reasons:
>
> (1) Our work contributes to the growing field of data-centric AI, which is increasingly recognized as an important part of the ML community as evidenced by recent ICLR (NeurIPS and ICML) papers, workshops (DMLR) and tutorials.
>
> (2) Schema matching is a challenging ML problem requiring reasoning along various dimensions (as outlined on **L98-L107**)  --- making it interesting from an ML methods and LLM reasoning perspective. Hence, developments of new ML methods to address this problem are fitting for ICLR (e.g. our self-improvement using synthetic in-context examples improves schema matching, but is generally applicable to optimizing multi-stage LLM programs).
>
>  (3) Improving schema matching is of interest to all ML researchers as it would increase the amount of data for the ML community to train and validate models (see **L82-87**)
>
> (4) We directly address recent calls in the ML community to develop methods for data harmonization/interoperability (Balagopalan et al., 2024; Gilbert et al., 2024). This highlights the recent interest in the ML community for methods like Matchmaker.
>
> We also believe that the dissemination of papers like Matchmaker within the ML community will spur greater ML innovation in schema matching — thereby improving methods and performance.
>
> **UPDATE:** We have updated the discussion on the relevance of schema matching to data-centric AI and its growing relevance in the ML community (L174-180)
>
> ----
> _Thank you for your help in improving our work. We hope these answer your points, please let us know if there are any remaining concerns!_

---

> ### Comment · Reviewer_fXFk · 2024-11-19
>
> Thanks for all the thoughtful comments and clarifications.
>
> I acknowledge the fact that you compare against other LLM-based approaches, and understand now the compositional approach as one of the main contributions of the paper. I have raised my score based on these clarifications.

---

> > ### Author Response · Authors · 2024-11-19
> >
> > Dear ``Reviewer fXFk``
> >
> > We sincerely appreciate your thoughtful engagement and are pleased that our responses have addressed your concerns, leading to an increased score.
> >
> > Given that we have now resolved your concerns, we kindly request that you reconsider raising your score to a 6, making the paper an accept (rather than still reject). We believe that this adjustment would more accurately reflect the contributions and overall impact of our work.
> >
> > If there are any remaining concerns, we would be happy to engage further to ensure the paper meets your expectations.
> >
> > Thank you once again for your valuable feedback and consideration.
> >
> > Paper 7911 Authors

---

> > > ### Comment · Reviewer_fXFk · 2024-11-26
> > >
> > > Unfortunately, I still consider that the contributions of this paper are limited. I see now that you are using and designing an interaction with LLMs that is novel, and brings some new ideas forward (which is why I raised my score), but the scope of the problem is using off-the-shelve models to solve schema matching, which I believe is marginally below the acceptance threshold for ICLR.

---

> > > > ### Author Response · Authors · 2024-11-26
> > > >
> > > > Dear ``Reviewer fXFk``
> > > >
> > > > Thank you for your continued engagement and thoughtful feedback. We appreciate the opportunity to further clarify the following:
> > > >
> > > > (i) How Matchmaker goes beyond just off-the-shelf LLMs.
> > > >
> > > > (ii) The importance of the problem for the ML community
> > > >
> > > > (iii) Broader impact
> > > >
> > > > ----
> > > >
> > > > ## **(i) How Matchmaker goes beyond just off-the-shelf LLMs**
> > > >
> > > > We want to emphasize that Matchmaker goes far beyond simply "using an off-the-shelf LLM". Our experiments (detailed in Table 2, Sec. 5.1) show that off-the-shelf LLM baselines like ReMatch and LLM-DP, fail to performance-wise on real-world schema matching tasks due to their inability to address structural and semantic complexities. Hence, we need to go something more --- specifically the contributions of Matchmaker.
> > > >
> > > > We respectfully emphasize that Matchmaker addresses the limitations of approaches of off-the-shelf LLM approaches like ReMatch and LLM-DP:
> > > >
> > > > - **LLM-DP and ReMatch (Off-the-shelf LLMs)**: As outlined in Section 2 and Table 1, these methods rely on static, single-step applications of off-the-shelf LLMs. They lack iterative reasoning and any mechanism for dynamic optimization, making them unsuitable for scaling to complex schema matching challenges. In addition, they do not consider deployment time deferrals to humans. This results in poor performance as shown in the paper.
> > > >
> > > >
> > > > - **Matchmaker**: In contrast, Matchmaker introduces a compositional LLM system that introduces the following contributions necessary to improve performance:
> > > >
> > > > (C1) Multi-Stage Reasoning: Described in Sections 4.1–4.3, Matchmaker decomposes schema matching into stages of candidate generation, refinement, and confidence scoring, enabling iterative reasoning that captures structural and semantic complexities.
> > > >
> > > > (C2) Zero-Shot Self-Improvement: In Section 4.4, Matchmaker introduces a novel mechanism using synthetic in-context examples to systematically optimize the multi-stage LLM program without labeled data. This allows Matchmaker to adapt and improve itself dynamically, going far beyond the static use of off-the-shelf LLMs.
> > > >
> > > > -----
> > > >
> > > > ## **(ii) Importance of Schema Matching for ML**
> > > > Schema matching is vital for advancing ML, as discussed in Sections 1 and 3, particularly in increasing access to large amounts of data needed to train ML models. In more detail:
> > > >
> > > > - Expanding ML-Ready Data: Schema matching enables data harmonization, unlocking diverse datasets for ML model training and validation. This is critical for improving model performance and external validation.
> > > > - Real-World Challenges: Domains like healthcare require the integration of heterogeneous, hierarchical datasets (e.g., MIMIC-OMOP schema mapping, which took 500 hours of expert work). Matchmaker addresses these challenges by automating this labor-intensive process while maintaining high accuracy.
> > > > - Contributions to Data-Centric AI: Matchmaker aligns with the growing interest in data-centric AI in the ML community. In particular, recent ICLR workshops and papers have emphasized the importance of data-centric problems. Matchmaker directly addresses this upstream challenge by enabling schema matching, paving the way for harmonized datasets that benefit the entire ML community.
> > > >
> > > > ---
> > > >
> > > > ## **(iii) Broader Impact**
> > > > - Superior Performance - Empirical Impact: Matchmaker’s outperformance of baselines like ReMatch and LLM-DP across real-world benchmarks is shown empirically. This highlights the need for our compositional pipeline and optimization framework to improve performance.
> > > >
> > > > - General ML impact: Matchmaker’s innovations around self-improvement extend beyond schema matching to other multi-stage LLM tasks, offering broad applicability to real-world ML challenges.
> > > >
> > > > ----
> > > >
> > > > **_Based on the above points, how we go beyond off-the-shelf LLMs, as well as its impact, we hope we have clarified our contributions more effectively. We sincerely thank you for giving us this opportunity to better position our work and articulate its impact. We respectfully ask you to reconsider your score in light of this additional context._**
> > > >
> > > > Thank you again for your thoughtful feedback and consideration.
> > > >
> > > > Sincerely,
> > > >
> > > > Paper 7911 Authors

---

> > > > > ### Author Response · Authors · 2024-12-02
> > > > >
> > > > > Dear ``Reviewer fXFk``
> > > > >
> > > > > Thank you for your engagement during this process.
> > > > >
> > > > > We hope our detailed clarifications have addressed your concern that Matchmaker uses an off-the-shelf LLM.
> > > > >
> > > > > With this in mind, we ask that the reviewer might consider raising their score to better align with this clarity of how we go beyond off-the-shelf LLMs and the impacts of Matchmaker
> > > > >
> > > > > Should there be any leftover concerns, we are committed to making every effort to resolve any outstanding issues in the limited time remaining!
> > > > >
> > > > > Regards
> > > > >
> > > > > Paper 7911 Authors

---

### Author Response · Authors · 2024-11-18
**Response Overview**

We thank the Reviewers for their insightful and positive feedback, and their time during the review process!

We are encouraged that they found the schema matching problem to be "important" (``R-fXFk``), "fundamental" (``R-N216``), ‘relevant and interesting’(``R-N216``) , that "addresses a critical practical problem"(``R-JXrD``) with "significant implications for ML development" (``R-JXrD``). They highlighted our "novel technical approach" (``R-JXrD``) noting particularly that "LLMs for schema matching is novel" (``R-TBJ4``) and highlighting our "zero-shot learning capability through synthetic in-context examples" (``R-JXrD``), along with its "advantage over previous proposals" (``R-fXFk``). We are pleased they noted our "comprehensive empirical evaluation"(``R-JXrD``) with "strong quantitative results" (``R-JXrD``), including our "(very nice) ablation study to understand the impact of different strategies" (``R-fXFk``) and that the proposed approach "outperforms" (``R-N216``) in various settings.

We address specific questions and concerns below and highlight updates based on reviewer suggestions.

**We have uploaded the updated manuscript with the changes highlighted in blue.** We thank the reviewers for helping us improve the paper!

On the basis of our clarifications and updates, we hope we have addressed the Reviewers' concerns.

Thank you for your kind consideration!

Paper 7911 Authors

---

### Meta-Review · Area_Chair_CQio · 2024-12-21

**Metareview:**

This paper proposes Matchmaker, a schema matching technique using a self-improving compositional language model program. The reviewers agree the paper addresses an important problem, is well written, and provides thorough experiments. During the rebuttal, the authors made significant efforts to address the reviewer concerns. However, there are still remaining concerns on the contributions/scope and mapping correctness despite extensive discussions. There was also an additional discussion among the reviewers, and the consensus is that while the paper is interesting, the contributions are not novel and that there are still unresolved concerns. Summing things up, I think the paper can be improved and should go through another round of reviewing.

**Additional Comments On Reviewer Discussion:**

Main remaining concerns:
- Contributions/scope: using off-the-shelve models to solve schema matching is not novel enough (reviewer fXFk), and a large body of schema matching literature is disregarded (reviewer N216).
- Mapping correctness: reviewer TBJ4 has various concerns on the correctness of the mappings.

---

### Decision · Program_Chairs · 2025-01-22

Reject